# Modeling Content Creator Incentives on Algorithm-Curated Platforms

**Jiri Hron**[⋆,†]**, Karl Krauth**[⋆,◇]**, Michael I. Jordan**[◇]**, Niki Kilbertus**[♠]**, Sarah Dean**[♣]
[†]University of Cambridge, [◇]UC Berkeley, [♠]TU Munich & Helmholtz Munich, [♣]Cornell University

## Abstract

Content creators compete for user attention. Their reach crucially depends on algorithmic choices made by developers on online platforms. To maximize exposure, many creators adapt strategically, as evidenced by examples like the sprawling search engine optimization industry. This begets competition for the finite user attention pool. We formalize these dynamics in what we call an *exposure game*, a model of incentives induced by algorithms, including modern factorization and (deep) two-tower architectures. We prove that seemingly innocuous algorithmic choices—e.g., non-negative vs. unconstrained factorization—significantly affect the existence and character of (Nash) equilibria in exposure games. We proffer use of creator behavior models, like exposure games, for an (ex-ante) *pre-deployment audit*. Such an audit can identify misalignment between desirable and incentivized content, and thus complement post-hoc measures like content filtering and moderation. To this end, we propose tools for numerically finding equilibria in exposure games, and illustrate results of an audit on the MovieLens and LastFM datasets. Among else, we find that the strategically produced content exhibits strong dependence between algorithmic exploration and content diversity, and between model expressivity and bias towards gender-based user and creator groups.

## 1 Introduction

In 2018, Jonah Peretti (CEO, Buzzfeed) raised alarm when a Facebook main feed update started boosting junk and divisive content (Hagey & Horwitz, 2021). In Poland, the same update caused an uptick in negative political messaging (Hagey & Horwitz, 2021). Tailoring content to algorithms is not unique to social media. For example, some search engine optimization (SEO) professionals specialize on managing impacts of Google Search updates (Marentis, 2014; Dennis, 2016; Shahzad et al., 2020; Patil et al., 2021; Goodwin, 2021). While motivations for adapting content range from economic to socio-political, they often translate into the same operative goal: *exposure maximization*.

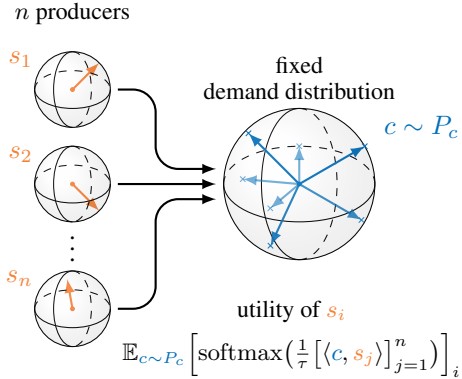

Figure 1: **Exposure game**. Items $s_i \in S^{d-1}$ placed to maximize exposure to consumers $c \sim P_c$.

We study how algorithms affect exposure-maximizing content creators. We propose a novel incentive-based behavior model called an *exposure game*, where producers compete for a *finite* user attention pool by crafting content ranked highly by a given algorithm (Section 1.1). When producers act strategically, a steady state—Nash equilibrium (NE)—may be reached, with no one able to unilaterally improve their exposure (utility). The content produced in a NE can thus be interpreted as what the algorithm implicitly incentivizes.

We focus on algorithms which model user preferences as an inner product of $d$-dimensional user and item embeddings, and rank items by the estimated preference. Section 2 presents theoretical results on the NE induced by these algorithms. We identify cases where algorithmic changes seemingly unconnected to producer incentives—e.g., switching from non-negative to unconstrained embeddings—determine whether there are zero, one, or multiple NE. The character of NE is also

affected by the level of algorithmic exploration. Perhaps counter-intuitively, we show that high levels of exploration incentivize broadly appealing content, whereas low levels lead to specialization.

In Section 3, we explore how creator behavior models can facilitate a pre-deployment audit. Such an audit could be particularly useful for assessing the producer impact of algorithmic changes, which is hard to measure by A/B testing for two important reasons: (1) producers cannot be easily randomized to distinct treatment groups, and (2) there is often a delay between deployment and content adaptation. Our hope is that this new style of auditing will enable detection of misalignment between the induced and desired incentives, and thus flag issues to either immediately address, or monitor in content filtering and moderation. For demonstration, we execute a pre-deployment audit on the MovieLens and LastFM datasets using the exposure game behavior model, and matrix factorization based recommenders. We find a strong dependence between algorithmic choices like embedding dimension and level of exploration, and properties of the incetivized content such as diversity (confirming our theory), and targeting of gender-based user and creator groups.

## 1.1 SETTING AND THE EXPOSURE GAME INCENTIVE MODEL

We assume there is a *fixed* recommender system trained on past data, and a *fixed* population of users (*consumers*). Together, these induce a *demand distribution* $P_c$ which represents typical traffic on the platform over a predefined period of time. Content is created by $n \in \mathbb{N}$ *producers* who try to maximize their expected exposure (*utility*). Denoting consumers by $c \sim P_c$, an item created by the $i^{\text{th}}$ producer by $s_i$ (*strategy*), $s := (s_i)_{i \in [n]}$, and $s_{\backslash i} := (s_j)_{j \neq i}$, we define (expected) *exposure* as the proportion of the "user attention pool" captured by the $i^{\text{th}}$ producer

$$u_i(s) = u_i(s_i, s_{\backslash i}) := \mathbb{E}_{c \sim P_c} \left[ \mathbb{1}\{c \text{ is exposed to } s_i\} \right] \overset{\star}{=} \mathbb{E}_{c \sim P_c}[p_i(c)], \qquad (1)$$

with $p_i(c) \geq 0$ the probability that the algorithm exposes $c$ to $s_i$ rather than any $s_{\backslash i}$. As common in game theory, we can extend from deterministic single item strategies to stochastic multi-item strategies $s_i \sim P_i$ for some distribution $P_i$. This extension is discussed in more detail in Section 2.

The assumption that $\mathbb{E}[\mathbb{1}\{c \text{ is exposed to } s_i\}] \overset{\star}{=} \mathbb{E}[p_i(c)]$ does not explicitly model interactions not mediated by the algorithm (e.g., YouTube videos linked to by an external website). This may be a reasonable approximation for infinite feed platforms (e.g., Twitter, Facebook, TikTok) where most consumers scroll through items in the algorithm-defined order, and search engines (e.g., Google, Bing) where first-page bias is well documented (Craswell et al., 2008). While similar assumptions are common in the literature (e.g., Li et al., 2010; Chen et al., 2019; Ben-Porat et al., 2020; Curmei et al., 2021), alternative interaction models are an important future research direction.

Unlike previous work (Section 1.2), we focus on the popular class of factorization-based algorithms. These models rank items by a score estimated by the inner product of user and item embeddings $c, s_i \in \mathbb{R}^d$. The larger this score, the higher the probability of exposure, which we model as

$$p_i(c) = \frac{\exp(\tau^{-1}\langle c, s_i\rangle)}{\sum_{i'=1}^n \exp(\tau^{-1}\langle c, s_{i'}\rangle)} = \text{softmax}\left(\left[\tau^{-1}\langle c, s_{i'}\rangle\right]_{i'=1}^n\right)_i, \qquad (2)$$

where $\tau \geq 0$ is a temperature parameter which controls the spread of exposure probabilities over the top scoring items. When $\tau = 0$ (i.e., hardmax), these probabilities correspond to top-1 recommendation or absolute first-position bias. Taking $\tau > 0$ models the effects of ranked position, injected randomness for exploration, and can partially adjust for user randomness and other factors which make top-ranked items receive more but not all of the traffic. While an approximation in some settings, Equation (2) has been directly used, e.g., by YouTube (Chen et al., 2019). We emphasize that we make no assumption on how the embeddings are obtained. Our conclusions thus apply equally to classical matrix factorization and deep learning-based systems.

We are now ready to formalize *exposure games*, an incentive-based model of creator behavior.

**Definition 1.** *An exposure game consists of an embedding dimension $d \in \mathbb{N}$, a demand distribution $P_c \in \mathcal{P}(\mathbb{R}^d)$, and $n \in \mathbb{N}$ producers, each of whom chooses a strategy $s_i \in S^{d-1} = \{v \in \mathbb{R}^d : \|v\| = 1\}$, to maximize their utility $u_i(s) = \mathbb{E}_{c \sim P_c}[p_i(c)]$ with $p_i(c)$ as in Equation (2) for a given $\tau \geq 0$.*

We restrict items $s_i$ to the unit sphere $S^{d-1}$. A norm constraint is necessary as otherwise exposure could be maximized by inflating $\|s_i\| \to \infty$, which is not observed in practice.[1] We distinguish

---

[1] Possibly due to the often finite rating scale, use of gradient clipping, and various forms of regularization.

*non-negative* games where all embeddings lie in the positive orthant; this includes algorithms ranging from TF-IDF, bag-of-words, to non-negative matrix factorization (Lee & Seung, 1999), topic models (Blei et al., 2003), and constrained neural networks (Ayinde & Zurada, 2017).

**Definition 2.** *A non-negative exposure game is an exposure game where the support of $P_c$ is restricted to the positive orthant, i.e., $P_c(\{c \in \mathbb{R}^d \colon c_j \geq 0 , \forall j \in [d]\}) = 1$.*

We assume all producers are rational, omniscient, and fully control placement of $s_i$ in $S^{d-1}$. These assumptions are standard in both machine learning and economics literature, including in the related facility location games (see Section 1.2). They often provide a good first order approximation, and an important basis for studying the subtleties of real-world behavior. Full control is perhaps the least realistic, since producers can modify content *features*, but they often do not know how these changes affect the content *embedding*. This assumption has a significant advantage though: it abstracts away an explicit model of producer actions (cf. the variety of SEO techniques). Appropriateness of rationality and complete information are then context-dependent; they may be respectively reasonable in environments where strong profit motives or user profiling tools are common. However, investigating alternatives to each of the above assumptions is an important direction of future work.

> **Box 1: How our assumptions map onto YouTube (YT) as an illustrative example.** On YT, a strategy $s_i$ is an embedding of a video, with creators able to produce multiple videos (mixed strategy $s_i \sim P_i$).
> **Rational behavior:** YT creators receive income proportional to their view numbers (Figure 2), which motivates exposure maximization. Most creators do not earn significant income, but the majority of traffic is driven by only a few popular and high-earning creators (Cheng et al., 2008). This motivates focus on these few producers and their strategic behavior.
> **Complete information and full-control.** YT
>
> 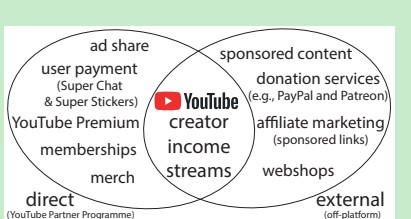
>
> Figure 2: YouTube revenue streams incentivizing exposure maximization (Ørmen & Gregersen, 2022).
>
> creators cannot *directly* manipulate the embeddings of their videos $s_i$, or observe the user embeddings. However, popular creators have a myriad of analytic tools at their hand, with information about views, demographics (e.g., gender, age, region), acquisition channels, drivers of engagement, competition and more. They can also observe and adopt behaviors of other creators. Taking the strong monetary incentives into account, motivated creators will actively optimize their exposure using trial-and-error, making complete information and full-control an imperfect yet not unreasonable model of their behavior.

## 1.2 RELATED WORK

Most relevant to our setup are works on the incentives of exposure-maximizing creators induced by recommender and retrieval systems (Ben-Porat et al., 2020; Raifer et al., 2017; Ben-Basat et al., 2017; Ben-Porat & Tennenholtz, 2018; Ben-Porat et al., 2019b;a). Interesting aspects of these works which we omit include (i) *repeated interactions* (Ben-Porat et al., 2020; Raifer et al., 2017; Ben-Porat et al., 2019b), (ii) *user welfare* (Ben-Porat et al., 2020; Ben-Basat et al., 2017; Ben-Porat & Tennenholtz, 2018; Ben-Porat et al., 2019a), and (iii) *incomplete information* (Raifer et al., 2017).

The most important distinction of our approach is that the above works constrain creators to a predefined finite item catalog. This excludes the popular *factorization-based* algorithms—ranging from standard matrix factorization (Koren et al., 2009) to (deep) two-tower architectures (Huang et al., 2013; Yi et al., 2019)—whose continuous embedding space translates into an *infinite* number of possible items. The only exception is (Ben-Porat et al., 2019a) where items are represented by $[0, 1]$ scalars, which is equivalent to the special case of two-dimensional non-negative exposure games. Continuous embedding spaces were recently studied in (Mladenov et al., 2020; Zhan et al., 2021), but neither studies producer incentives or competition. Mladenov et al. (2020) consider producers who decide whether to stay or leave the platform if their exposure is too low. Zhan et al. (2021) study design of recommender systems which optimize for *both* user and producer utility.

Concurrently but independently, Jagadeesan et al. (2022) study a model equivalent to *hardmax non-negative exposure games*, except the $\|s_i\| = 1$ constraint is replaced by a *production cost*, yielding

$u_i(s) = \mathbb{E}[p_i(c)] - \|s_i\|^\beta$ for some norm $\|\cdot\|$ and $\beta \geq 1$ (higher norm interpreted as higher quality). The authors investigate how the cost function influences the economic phenomena exhibited by NE, from formation of "genres" (multiple directions with non-zero probability), to the possibility of realizing positive profits (utility). In contrast, we investigate how NE depend on algorithmic and environmental factors (non-negativity, exploration, dependence of exposure on ranking), and propose an algorithmic audit which leverages the creator model. While taking $\beta \to \infty$ in the Jagadeesan et al.'s cost recovers our unit norm constraint, understanding the NE behavior at the limit remains a subject of future work (e.g., pure NE exist only in our setup). Our works are thus largely complementary.

Literature on adaptive behavior in the presence of a prediction algorithm is also relevant (Hardt et al., 2016; Kleinberg & Raghavan, 2020; Perdomo et al., 2020; Jagadeesan et al., 2021). The social impact and potential disparate effects of strategic adaptation have been analyzed in (Milli et al., 2019; Hu et al., 2019; Liu et al., 2020). Most relevant for us is a recent paper by Liu et al. (2022) which studies strategic adaptation in the context of *finite* resources (e.g., number of accepted college applicants). Unlike us, the authors assume a *single* score for each competitor, who can pay *cost* to improve it. A principal then *designs* a reward function which allocates the finite resource based on the scores, and the authors study how different choices affect various notions of welfare. The preliminary results on multi-dimensional scores (appendix B) assume the scores and individual improvements are *independent*, whereas our scores—$\langle c, s_i \rangle$ for each $c$—imply complex dependence and trade-offs.

Finally, our proposed methods for auditing recommender and information retrieval systems belong to a rapidly growing algorithm auditing toolbox. We focus on understanding producer incentives caused by a known algorithm. Thus, we complement prior work that aims to audit these systems based upon: the degree of consumer control (Curmei et al., 2021), fairness (Do et al., 2021), compliance with regulations (Cen & Shah, 2021), and dynamical behavior in simulations (Krauth et al., 2020; Lucherini et al., 2021) or deployed systems Haroon et al. (2022).

## 2 EQUILIBRIA IN EXPOSURE GAMES

This section presents theoretical results on incentives in exposure games. We focus on the impact of the recommender/information retrieval model on the competitive equilibria. Throughout, we find that one of the most important factors determining existence and character of equilibria is the temperature $\tau$ (see Equation (2)). We thus distinguish the *softmax* ($\tau > 0$) and the *hardmax* ($\tau = 0$) case.

In competitive settings, a key question is whether there are equilibria in which players are satisfied with their strategies, as otherwise there may be never-ending oscillation in search for better outcomes. We thus consider several *solution concepts* (i.e., definitions of equilibria) related to NE. A *pure NE* (PNE) is a point in strategy space $s^{\mathsf{NE}} \in (S^{d-1})^n$ where no player $i$ can increase their utility by unilaterally deviating from $s_i^{\mathsf{NE}} \in S^{d-1}$. In other words, no content producer can increase their exposure by modifying their content. *Mixed NE* (MNE) refer to the setting where players are allowed to choose randomized (*mixed*) strategies $P_i \in \mathcal{P}(S^{d-1})$. Rather than selecting a single piece of content, a creator following a mixed strategy samples $s_i \sim P_i$. Alternative interpretation is that producers create multiple items, splitting their time/budget proportionally to the $P_i$-probabilities.

In later sections, we explore the weaker solution concepts of $\epsilon$-NE, *local NE* (LNE), and their combination $\epsilon$-LNE. An $\epsilon$-NE is an approximate NE where no producer can unilaterally increase their utility by more than $\epsilon$ (NE are "0-NE"). LNE are analogous to local optima: points where no player benefits from small deviations from their strategy. The approximate and local perspectives are relevant when deploying local search algorithms to find NE numerically (Section 3).

Exposure games are symmetric, meaning that any permutation of strategies forming an equilibrium produces another equilibrium. Our statements on the existence and uniqueness of equilibria hold up to player permutation. All proofs for the results in this section are presented in the appendix.

### 2.1 PURE AND MIXED NASH EQUILIBRIA

We begin by characterizing the existence of pure and mixed NE in general exposure games.

**Theorem 1.** *Every exposure game has at least one* mixed *Nash equilibrium.*

A key property of *softmax* games is that the utilities $u_i$ are continuous in $s$. This, and the compactness of the strategy space $S^{d-1}$, guarantees existence of MNE (Glicksberg, 1952, section 2). In the *hardmax* case ($\tau = 0$), we can show that MNE are guaranteed to exist through a direct application

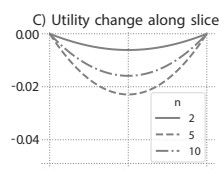 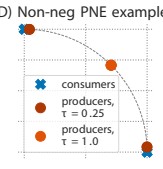 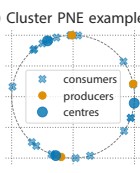

Figure 3: **A)** A game with no PNE (see the proof of Theorem 2). A PNE would exist if the strategy space was *convex*, and utility *quasi-concave* (Fan, 1952). B) and C) demonstrate lack of quasi-concavity even if we allow $\|s_i\| \leq 1$: **B)** $n-1$ producers at midpoint, $s_1$ along slice $\lambda c_1 + (1-\lambda)c_2$ (dashed line); **C)** Change in utility along the slice in B) demonstrates lack of quasi-concavity. **D)** A non-negative game with very different PNE depending on $\tau$. **E)** PNE with "protective positioning."

of proposition 4 due to Simon (1987). The producer utilities $u_i$ are not differentiable in the hardmax case though, which means we cannot use gradient information to find NE as in the softmax case. The only procedure we know for finding NE in hardmax games requires solving the hitting set problem which is NP-complete (Dasgupta et al., 2008). See Appendix B for further discussion.

We now turn to existence of *pure* NE, which is the setting where creators strategically design a *single* piece of content. Unlike MNE, PNE are not guaranteed to exist even in the softmax case.

**Theorem 2.** *PNE need not exist in either the hardmax ($\tau = 0$) or softmax ($\tau > 0$) exposure games.*

Figure 3A illustrates the non-existence result. The counter-example holds even for $n = 2$ players and planar ($d = 2$) strategies. A reader familiar with classic PNE results may ask if PNE would appear if we relaxed the $S^{d-1}$ strategy space to the *convex* $B^d = \{v \colon \|v\| \leq 1\}$ (Glicksberg, 1952; Debreu, 1952; Fan, 1952). This is not true as the exposure utility is not quasi-concave (Figure 3B&C).

We now move to *non-negative* exposure games (Definition 2). For $n = d = 2$, non-negative hardmax exposure games are equivalent to Hotelling games (Hotelling, 1929), and more generally to facility location games on a line (Ben-Porat et al., 2019a; Procaccia & Tennenholtz, 2013). The next proposition lists several special cases in which we understand existence and character of PNE.

**Proposition 1.** *A PNE always exists in $n = d = 2$ non-negative hardmax games, but may not without non-negativity or when $d > 2$. For $n = 2$ non-negative softmax games with $\hat{c} \coloneqq \frac{1}{n}(1 - \frac{1}{n})\mathbb{E}[c] \neq 0$, the only possible PNE is $s_1 = s_2 = \bar{c}$ with $\bar{c} \coloneqq \hat{c}/\|\hat{c}\|$ (independently of $d$), but a PNE may not exist. When $n > 2$, non-negative softmax games can have a PNE other than $s_1 = \cdots = s_n = \bar{c}$.*

Figure 3D illustrates a 4-player non-negative exposure game. Depending on the temperature, we observe either the collapsed $s_i = \bar{c}$ (large $\tau$), or what we term "protective positioning" (small $\tau$). In Figure 3D, players place their strategies *between* a consumer and the next closest producer. Figure 3E illustrates protective positioning for a higher number of consumers and $n = 3$. Here, consumers are roughly clustered around three centers (blue dots). The producer strategies are close to these centers, but again offset towards the most contested consumers.

## 2.2 $\epsilon$-NASH EQUILIBRIA

While existence of NE is not guaranteed, the situation changes when we adopt the weaker solution concept of $\epsilon$-NE, in which no producer can unilaterally increase their utility by more than $\epsilon$.

The existence and character of such equilibria strongly depends on the temperature $\tau$. When $\tau = \infty$, exposure is equally likely $p_i(c) = \frac{1}{n}$ for all $i$ and $c$ regardless of the adopted strategies. Thus, every strategy profile is an NE. Considering a sequence of increasing $(\tau_i)_{i \geq 1}$, we can therefore argue that the limit of any convergent sequence of NE indexed by $\tau$ is a NE at $\tau = \infty$. Interestingly, Theorem 3 shows that a sufficiently large but *finite* $\tau > 0$ is sufficient for existence of $\epsilon$-(P)NE. The result is constructive, showing that the $\epsilon$-PNE is parallel to the average consumer embedding.

**Theorem 3.** *For any $\epsilon > 0$ and $P_c \in \mathcal{P}(\mathbb{R}^d)$ with compact support and $\mathbb{E}[c] \neq 0$, $\exists \tau_0 > 0$ s.t. $s_1 = \ldots = s_n = \bar{c}$ is an $\epsilon$-PNE for all $\tau \geq \tau_0$. Moreover, for all $\tau \geq \tau_0$, the smallest $\epsilon_\tau$ for which $\bar{c}$ is an $\epsilon_\tau$-PNE satisfies $\epsilon_\tau \leq \frac{\epsilon}{\tau}$. If also $\epsilon < \|\hat{c}\|$, then the set of better-responses to $\bar{c}$*

$$\Psi(\bar{c}) \coloneqq \{v \in S^{d-1} \colon u_1(v, \bar{c}, \ldots, \bar{c}) \geq u_1(\bar{c}, \bar{c}, \ldots, \bar{c})\}, \tag{3}$$

*is a subset of $B_\delta^d(\bar{c}) = \{v \colon \|v - \bar{c}\| \leq \delta\}$ with $\delta = 2\epsilon/(\|\hat{c}\| - \epsilon)$, and $\delta \to 0$ as $\tau \to \infty$.*

This result shows that all $\epsilon$-improvements concentrate near the consumer average $\epsilon$-PNE as $\tau \to \infty$. Additionally, the "consumer symmetry" $\|\hat{c}\| = \frac{1}{n}(1 - \frac{1}{n})\|\mathbb{E}[c]\|$ determines how quickly $\delta \to 0$. When consumers are spread approximately symmetrically w.r.t. the origin, the degenerate equilibrium appears only for large $\tau$. However, smaller $\tau$ are sufficient for more directionally concentrated $P_c$. A high number of producers also slows the concentration as the appeal of $u_i(\bar{c}, \ldots, \bar{c}) = \frac{1}{n}$ decreases with $n$. We conclude with a corollary based on our development so far.

**Corollary 1.** *There is a fixed $\epsilon_0 > 0$ and a demand distribution $P_c$ which—depending on the chosen $\tau$—induce zero, one, multiple, or infinitely many $\epsilon$-NE for all $\epsilon \leq \epsilon_0$.*

Corollary 1 underscores the sensitivity of exposure games to the temperature parameter $\tau$, with uniformly homogeneous content at one end (high $\tau$), and potentially persistent oscillation behavior in competition when no NE exist (low $\tau$). A higher $\tau > 0$ can be a result of *algorithmic exploration* (Chen et al., 2019; Cesa-Bianchi et al., 2017; Lattimore & Szepesvári, 2020), which is provably necessary for optimal performance in *static* environments (Lattimore & Szepesvári, 2020). In contrast, our results show that in environments with *strategic* actors, exploration may incentivize content which is uniform and broadly appealing rather than diverse.

This may contradict the intuition that more exploration should lead to greater content diversity due to the higher exposure of niche content. One way to understand this result is the tension between randomization and the ability of niche creators to reach their audience: producers may be discouraged from creating niche content when the algorithm is exploring too much ($\tau$ high), and encouraged to mercilessly seek and protect their own niche when the algorithm performs little exploration ($\tau$ low). When the algorithm captures user preferences well, exploration is typically thought of as having negative impact on user experience through immediate reduction in quality of service as a result of suboptimal recommendations. However, the above results show secondary long-term effects.

### 2.3 LOCAL NASH EQUILIBRIA

In a local NE, each $s_i$ is optimal on some of its neighborhood within the embedding space. Sometimes motivated as a form of bounded rationality, LNE can often be found by local search algorithms (e.g., Mazumdar et al., 2019). Since our motivation in studying exposure games is ultimately better system understanding and audits, we are particularly interested in these algorithmic benefits.

Practical first-order algorithms for identifying LNE operate analogously to gradient descent, implying they may terminate in *critical points* that are not LNE. Unlike NE, critical points always exist.

**Proposition 2.** *Every $\tau > 0$ exposure game with $\mathbb{E}[c] \neq 0$ has a critical point at $s_1 = \ldots = s_n = \bar{c}$.*

As we have seen, $s_1 = \ldots = s_n = \bar{c}$ may be an equilibrium (Proposition 1). To distinguish LNE from mere critical points, we use the Riemannian second derivative test, treating $S^{d-1}$ as a Riemannian submanifold of $\mathbb{R}^d$ as usual. For background, see (Boumal, 2022, sections 3 & 5).

**Definition 3** (Boumal, 2022, lemma 5.41)**.** *A point $s$ in strategy space satisfies the* second derivative test *if $\forall i$ (1) the* Riemannian gradient $(I - s_i s_i^\top)\nabla_{s_i} u_i(s)$ *is zero, and (2) the* Riemannian Hessian

$$(I - s_i s_i^\top)\left[\nabla_{s_i}^2 u_i(s)\right](I - s_i s_i^\top) - \langle s_i, \nabla_{s_i} u_i(s)\rangle(I - s_i s_i^\top),$$

*is strictly negative definite in the subspace perpendicular to $s_i$.*

This condition is sufficient but not necessary for a critical point to be an LNE. LNE which satisfy Definition 3 are termed *differentiable* NE (Ratliff et al., 2016; Balduzzi et al., 2018). The distinction is similar to that between the flat minimum of $x^4$ at zero the more well-behaved $x^2$.

## 3 PRE-DEPLOYMENT AUDIT OF STRATEGIC CREATOR INCENTIVES

Beyond regularly retraining on new data, online platforms continuously roll out algorithm updates. While A/B testing can detect changes in user metrics, like satisfaction or churn, prior to the full-scale deployment (Tang et al., 2010; Hohnhold et al., 2015; Xu et al., 2015; Gordon et al., 2019), assessing the impact on content producers is comparatively harder due to the longer delay between the roll-out and corresponding content adaptation. Furthermore, since producers cannot be easily assigned to distinct treatment groups without limiting their content to only a subset of consumers, modern A/B testing methods must eschew making causal statements about producer impact (Nandy et al., 2021; Ha-Thuc et al., 2020; Huszár et al., 2022). Undesirable results including promulgation of junk and abusive content then have to be addressed via *post-hoc* measures like content filtration and moderation.

Figure 4: **Clustering of strategic producers depends on the exploration level** $\tau$. A cluster is a set of points whose Euclidean distances from one another are less than $10^{-5}\sqrt{d}$. As Theorem 3 predicts, large $\tau$ (e.g., more exploration) leads to higher concentration, i.e., creating content which appeals to *more* users. **Left:** MovieLens. **Right:** LastFM. See Section 3.2 for more discussion.

A tool for *ex-ante* (pre-deployment) assessment of producer impact could thus limit the harm experienced by users, moderators, and other affected parties. We demonstrate how to utilize a creator behavior model for this purpose, using the exposure game as a concrete example. The incorporation of factorization-based algorithms in exposure games allows us to use real-world datasets and rating models. While exposure games have limitations as a behavior model, we believe our experiments provide a useful illustration of the insights the proposed audit can offer to platform developers.

### 3.1 SETUP

We use the `MovieLens-100K` and `LastFM-360K` datasets (Harper & Konstan, 2015; Bertin-Mahieux et al., 2011; Shakespeare et al., 2020), implement our code in Python (van Rossum & Drake, 2009) and rely on `numpy` (Harris et al., 2020), `scikit-surprise` (Hug, 2020), `pandas` (pandas development team, 2020), `matplotlib` (Hunter, 2007), `jupyter` (Kluyver et al., 2016), `reclab` (Krauth et al., 2020), and `JAX` (Bradbury et al., 2018) packages to fit probabilistic (PMF; Mnih & Salakhutdinov, 2007) and non-negative (NMF; Lee & Seung, 1999) matrix factorization. The models are trained to predict the user ratings (centered in the PMF case). To select regularization and learning rate, we performed a two-fold 90/10 split cross-validation separately on each dataset. The tuned recommenders were then fit on the full dataset, and the resulting user embeddings, $\{c_j\}_{j\in[m]} \subset \mathbb{R}^d$, were used to construct the demand distribution $P_c = \frac{1}{m}\sum_j \delta_{c_j}$, and evaluate the recommendation probabilities $p_i(c)$. Details in Appendix C.1.

The only algorithm for finding NE in hardmax exposure games we know has exponential worst-case complexity. We thus focus on the softmax case. While search for general *mixed* NE is possible in special cases (Fudenberg & Kreps, 1993; Kaniovski & Young, 1995; Benaïm & Hirsch, 1997), we are not aware of any technique applicable to $n$-player exposure games. We therefore focus on *pure* $\epsilon$-LNE (Section 2.3), where each producer creates a single new item. We employ simple gradient ascent (Singh et al., 2000; Balduzzi et al., 2018, see Appendix C.2 for comparison with gradient descent) *combined with reparametrization* $s_i = \theta_i/\|\theta_i\|$ for each producer, where we iteratively update $\theta_{i,t} = \theta_{i,t-1} + \alpha\nabla_{\theta_{i,t-1}}u_i(s_{i,t-1}, s_{\setminus i,t-1})$ for shared step size $\alpha > 0$, and

$$\nabla_{\theta_i}u_i(s) = \tfrac{1}{\tau\|\theta_i\|_2}(I - s_is_i^\top)\,\mathbb{E}[p_i(c)(1-p_i(c))c] = \tfrac{1}{\|\theta_i\|_2}(I - s_is_i^\top)\nabla_{s_i}u_i(s)\,. \tag{4}$$

Equation (4) shows the update direction is parallel to the Riemannian gradient of $u_i(s)$ w.r.t. $s_i \in S^{d-1}$ (Section 2.3). We also experimented with the related Riemannian gradient ascent optimizer (Boumal, 2022), but abandoned it after (predictably) observing little qualitative difference. We note that the local updates themselves define *better-response dynamics* linked to iterative minor content changes; investigation of their relation to real-world producer behavior is an interesting future direction.

We investigate the sensitivity of the incentivized content to the: (i) *rating model* $\in$ {PMF, NMF}, (ii) *embedding dimension* $d \in \{3, 50\}$, and (iii) *temperature* $\log_{10}\tau \in \{-2, -1, 0\}$. We further vary the number of producers $n \in \{10, 100\}$ to examine scenarios with different producer to consumer ratios (user count is fixed to the full 943 for MovieLens, and 13,698 for LastFM). The above values were selected in a preliminary sweep as representative of the effects presented below. For every setting, we used five random seeds for initialization of the recommender (affects $P_c$), and for each ran the gradient ascent algorithm 10x to identify possible $\epsilon$-LNE. We applied early stopping when $\ell^2$-change in parameters between iterations dipped below $10^{-8} \cdot \sqrt{d}$; the number of iterations was set to 50K so convergence was achieved for *every* run. We only report runs where the second-order Riemannian test from Section 2.3 did not rule out an $\epsilon$-LNE. Additional results, including those where the Riemannian test was conclusive, are in Appendix C.3.

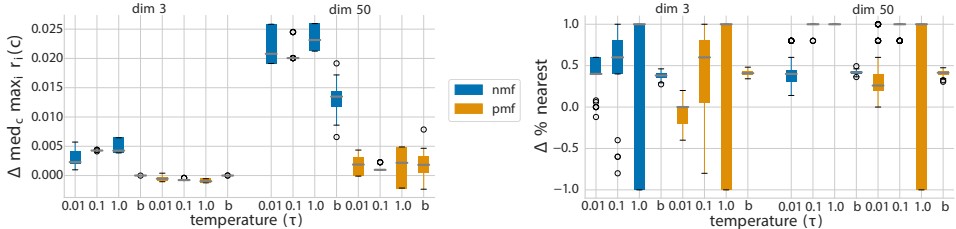

Figure 5: **Targeting of incentivized content by gender** on MovieLens. **Left:** Difference between $\text{median}_{c \in G}\{\max_{i \in [n]} \bar{r}_i(c)\}$ for men and women (group $G$), with $\bar{r}_i(c)$ the normalized rating (cosine similarity between $c$ and the strategic $s_i$). Positive values imply bias towards men (higher median). Note the higher bias when $d = 50$ (more expressive algorithm); especially NMF incentivizes more biased content relative to the pre-adaptation baseline 'b'. **Right:** Difference in proportions of $s_i$ with best (normalized) rating by women/men. Positive values imply bias towards men (more items best-rated by men). Bias again more pronounced at $d = 50$. See Section 3.2 for more discussion.

## 3.2 RESULTS

**Emergence of clusters with growing $\tau$.** Theorem 3 shows that producers concentrate around $\bar{c} = \mathbb{E}[c]/\|\mathbb{E}[c]\|$ for sufficiently high $\tau$. Figure 4 corroborates the result on both MovieLens and LastFM, with the concentration happening already at $\tau = 1$ regardless of the embedding dimension $d$ and producer count $n$. We also see that lower $\tau$ can lead to "local clustering" where only few producers converge onto the same strategy. We hypothesize that the *simultaneous* local updates of the consumers create "attractor zones" where close-by producers collapse onto each other; they will remain collapsed henceforth due to equality of their gradients (by symmetry). Theorem 3 does tell us collapse is to be expected for high $\tau$, and it is possible that a local version of the result with more than one clusters is true for intermediate values of $\tau$. This highlights how crucial the *algorithmic choice* of $\tau$ is for the induced incentives within our model.

**Targeting of incentivized content by gender.** The MovieLens dataset contains binarized user gender information. In Figure 5, we examine targeting of incentivized content on women and men. To do so, we employ aggregate statistics of *predicted* ratings. While predicted ratings may differ from actual user preferences, they do determine recommendations and thus *user experience*. To help disentangle effect of exposure maximization, we also include statistics based on the original item locations (labeled by 'b'), i.e., the content created before producers adapt to the recommender. Since the baseline embeddings need not satisfy the unit norm constraint (see Definition 1), we measure *normalized* ratings $\bar{r}_i(c) := \langle c, s_i \rangle / \|c\|\|s_i\|$ to facilitate comparison. The normalization also alleviates the known issue of varying interpretation of ranking scales between users (Lynch Jr et al., 1991).

In Figure 5 (left), $\text{median}_{c \in \text{men}}\{\max_i \bar{r}_i(c)\} - \text{median}_{c \in \text{women}}\{\max_i \bar{r}_i(c)\}$ measures if the incentivised content is predicted to appeal to women/men; Figure 5 (right) shows the fraction of creators incentivised to target women/men: $\frac{1}{n}\sum_{i=1}^{n} \mathbb{1}\{\text{argmax}_c \bar{r}_i(c) \in \text{men}\} - \mathbb{1}\{\text{argmax}_c \bar{r}_i(c) \in \text{women}\}$. Positive values signify content crafted for male audience (users are 71% male). Higher embedding dimension results in more bias, presumably due to the larger model expressivity, and thus enables more fine-grained targeting. NMF consistently incentivizes more biased content.

**Association between incentivized content and creator gender.** Platform developers may want to know if some creators are being disadvantaged (Chokshi, 2017; Farokhmanesh, 2018; Rodriguez, 2022). While solutions were proposed in the static case (e.g, Beutel et al., 2019; Wang et al., 2021), understanding if the algorithm (de)incentivizes content by particular creator groups may limit future harm. In Figure 6, we measure the difference between the *proportion of* (left) and the *median distance to* (right) *baseline* creator embeddings (learned by the recommender before strategic adaptation), within increasingly large neighborhoods of each strategic $s_i$. Since the baseline embeddings need not be unit norm, we use the cosine distance to define the neighborhoods.

Starting with the proportion (left), higher embedding dimension (more flexible model) incentivizes content more typical of male artists. This may be related to the higher prevalence of men in LastFM, combined with training by average loss minimization. The gender imbalance also explains why the proportion (left) stabilizes at a positive value, whereas the median distance (right) reverts to zero, as the number of considered neighbors grows. The bias is also related to the choice of rating model, where especially PMF at high temperatures results in significant advantage for male artists.

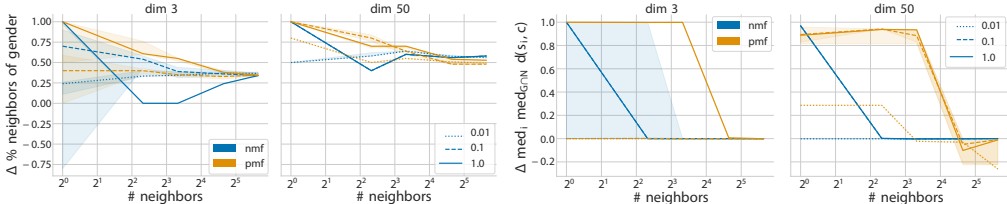

Figure 6: **Incentivized content and creator gender** on LastFM. Quantifying relative difficulty of strategic adaptation for female and male content creators, Uses baseline creator embeddings (and associated gender), and their cosine distance from strategic embeddings. **Left:** Difference between fractions of male and female creators in increasingly large neighborhood of each strategic item. Values above zero imply bias towards male producers. Higher embedding dimension (model expressivity) again results in larger bias. The bias also seems to be larger for higher $\tau$ and for the PMF rating model. **Right:** Difference between median cosine distance to female and male creators within increasingly large neighborhood of each strategic item. Values above zero imply bias towards male producers. Higher bias is again associated with higher embedding dimension and the PMF rating model, but the impact of temperature $\tau$ is less pronounced. See Section 3.2 for more discussion.

## 4 DISCUSSION

From social media and streaming to Google Search, many of us interact with recommender and information retrieval systems every day. While the core algorithms have been developed and analyzed years ago, the socio-economic context in which they operate received comparatively little attention in the academic literature. We make two main contributions: (a) we define *exposure games*, an incentive-based model of content creators' interactions with real-world algorithms including the popular matrix factorization and two-tower systems, and (b) we formulate a *a pre-deployment audit* which employs a model of creator behavior to identify misalignment between incentivized and desirable content.

Our main theoretical contributions focus on the properties of Nash equilibria in exposure games. We found that seemingly innocuous *algorithmic* choices like temperature $\tau$, embedding dimension $d$, or a non-negativity constraint on embeddings can have serious impact on the induced incentives. For example, high $\tau$ incentivizes uniform broadly appealing content, whereas low $\tau$ motivates targeting smaller consumer groups. Since higher $\tau$ is often linked to exploration, which is necessary for optimal performance in static settings (e.g., Lattimore & Szepesvári, 2020), this result highlights the importance of considering the socio-economic context in algorithm development.

Our producer model has several limitations from assuming rationality, complete information, and full control, to taking the skill set of each producer to be the same, their utility to be linear in total exposure, and ignoring algorithmic diversification of recommendations. We also consider the attention pool as fixed and finite, neglecting the problematic reality of the modern attention economy, where online platforms constantly struggle to increase their user numbers and daily usage (Covington et al., 2016; Williams, 2018; Bhargava & Velasquez, 2021). Our theoretical understanding is incomplete as, e.g., our understanding of the influence of constraining embeddings to be non-negative is limited to the two-dimensional case. The empirical evaluation of our behavior model is hindered by the lack of academic access to the almost exclusively privately owned platforms (Greene et al., 2022).

Due to their sizable influence on individuals, societies, and economy (Milano et al., 2020), information and recommender systems are of critical importance from an ethical and societal perspective. While we hope that a better understanding of the incentives these algorithms create will mitigate their negative social consequences, this also entails risks. Perhaps the most important is the possibility of employing an optimizer such as the one in Section 3 to game a real-world algorithm. This is especially relevant to the current debate about transparency (e.g., Sonboli et al., 2021; Rieder & Hofmann, 2020; Sinha & Swearingen, 2002), and the proposal to (partially) open-source the Twitter code base (Knight, 2022). Due to the aforementioned limitations, we also caution against treating the predictions of our incentive-based behavior model as definitive, especially given the significant complexity of many real-world algorithms and the environments in which they operate.

Going forward, we want to deepen our understanding of exposure games, and make pre-deployment audits a practical addition to the algorithm auditing toolbox. We hope this research enriches the debate about online platforms by a useful perspective for thinking about harms, (over)amplification, and design of algorithms with regard to the relevant incentives of the involved actors.

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

## A  PROOFS

**List of abbreviations:**

- s.t. = such that
- a.s. = almost surely
- l.h.s. = left-hand side
- r.h.s. = right-hand side
- w.l.o.g. = without loss of generality

### A.1  STAND-ALONE STATEMENTS

**Theorem 2.** *PNE need not exist in either the hardmax ($\tau = 0$) or softmax ($\tau > 0$) exposure games.*

*Proof of Theorem 2.* **(I) Hardmax:** Take $n = d = 2$ and $P_c = \frac{1}{3}\sum_{j=1}^{3}\delta_{c_j}$ where the angle between any $c_j \neq c_k$ is $\frac{2\pi}{3}$. Assume $s = (s_1, s_2)$ is a PNE. W.l.o.g. $c_1 = \mathrm{argmax}_j \langle s_1, c_j \rangle$. Then there is $s_2$ on the geodesic connecting $c_2$ and $c_3$ which has higher dot product with both $c_2$ and $c_3$ than $s_1$. Hence $u_2(s) \geq {}^2\!/_3$ by the assumption that $s$ is a PNE. The same argument implies $u_1(s) \geq {}^2\!/_3$. This is a contradiction since $\sum_i u_i(s) = 1$ by definition of the exposure utility.

**(II) Softmax:** Let $n = d = 2$, and $P_c = \frac{1}{3}(2\delta_{e_1} + \delta_{e_2})$ where $e_1 = [1, 0]^\top$ and $e_2 = [0, 1]^\top$. By Proposition 1, we know that the only possible PNE is $s_1 = s_2 = \bar{c} \propto \mathbb{E}[c] = [2, 1]/3$, where both

players enjoy $u_1(s) = u_2(s) = \frac{1}{2}$. Let $s_1' = (\bar{c} + \epsilon e_1)/\|\bar{c} + \epsilon e_1\|$ for some $\epsilon > 0$. As $\tau \to 0$, $u_1(s_1', \bar{c}) \to \frac{2}{3}$ by continuity. Hence $\exists \tau_0 > 0$ s.t. $s_1 = s_2 = \bar{c}$ is not a PNE for all $\tau < \tau_0$. $\qquad \square$

**Theorem 3.** *For any $\epsilon > 0$ and $P_c \in \mathcal{P}(\mathbb{R}^d)$ with compact support and $\mathbb{E}[c] \neq 0$, $\exists \tau_0 > 0$ s.t. $s_1 = \ldots = s_n = \bar{c}$ is an $\epsilon$-PNE for all $\tau \geq \tau_0$. Moreover, for all $\tau \geq \tau_0$, the smallest $\epsilon_\tau$ for which $\bar{c}$ is an $\epsilon_\tau$-PNE satisfies $\epsilon_\tau \leq \frac{\epsilon}{\tau}$. If also $\epsilon < \|\hat{c}\|$, then the set of better-responses to $\bar{c}$*

$$\Psi(\bar{c}) := \left\{ v \in S^{d-1} : u_1(v, \bar{c}, \ldots, \bar{c}) \geq u_1(\bar{c}, \bar{c}, \ldots, \bar{c}) \right\}, \tag{3}$$

*is a subset of $B_\delta^d(\bar{c}) = \{v : \|v - \bar{c}\| \leq \delta\}$ with $\delta = 2\epsilon/(\|\hat{c}\| - \epsilon)$, and $\delta \to 0$ as $\tau \to \infty$.*

*Proof of Theorem 3.* We w.l.o.g. focus on the defection strategies for $s_1$. By the mean-value theorem

$$\Delta := u_1(s_1, \bar{c}, \ldots, \bar{c}) - u_1(\bar{c}, \ldots, \bar{c}) = \langle g_1', s_1 - \bar{c} \rangle,$$

where $g_1' = \nabla_{s_1'} u_1(s_1', \bar{c}, \ldots, \bar{c})$ for some $s_1'$ on the *line* connecting $s_1$ and $\bar{c}$. While the rigorous argument below relies on a few technicalities, the main idea is simple: as $\tau \to \infty$, $\tau \cdot g_1' \to \hat{c} = \frac{1}{n}(1 - \frac{1}{n}) \mathbb{E}[c]$ *uniformly* over $s_1 \in S^{d-1}$ (Lemma 1), and thus $\tau \cdot \Delta \approx \langle \hat{c}, s_1 - \bar{c} \rangle \leq \|\hat{c}\|(1 - 1) = 0$.

**Lemma 1.** $\lim_{\tau \to \infty} \sup_{s_1 \in S^{d-1}} \|\tau \cdot g_1' - \hat{c}\| \to 0$.

*Proof of Lemma 1.* Since $\text{supp}(P_c)$ is compact by assumption, and $\tau \cdot g_1' = \mathbb{E}[p_1(c)(1 - p_1(c))c]$, all we need is $p_1(c)(1 - p_1(c)) \to \frac{1}{n}(1 - \frac{1}{n})$ *pointwise* in $c$ (dominated convergence), and *uniform* over $s_1' \in B^d = \{v : \|v\| \leq 1\}$ (mean-value theorem yields $s_1'$ on the *line* connecting $s_1$ with $\bar{c}$). As

$$p_1(c) = \frac{\exp(\tau^{-1}\langle c, s_1' \rangle)}{\exp(\tau^{-1}\langle c, s_1' \rangle) + (n-1)\exp(\tau^{-1}\langle c, \bar{c} \rangle)},$$

is monotonic in $\tau^{-1}\langle c, s_1' \rangle$, and $\sup_{s_1' \in B^d} \langle c, s_1' \rangle = \|c\| < \infty$ by compactness, $p_1(c)(1 - p_1(c))$ will converge to $\frac{1}{n}(1 - \frac{1}{n})$ uniformly over $B^d$ by continuity of the exponential function at zero. $\qquad \square$

For any given $\epsilon > 0$, Lemma 1 can be combined with

$$\tau \cdot \Delta \leq \langle \hat{c}, s_1 - \bar{c} \rangle + \|\tau \cdot g_1' - \hat{c}\| \|s_1 - \bar{c}\|,$$

where $\langle \hat{c}, s_1 - \bar{c} \rangle \leq 0$ for all $s_1 \in S^{d-1}$ by $\bar{c} = \hat{c}/\|\hat{c}\|$, to obtain $\Delta < \varepsilon$ for a sufficiently large $\tau$. In particular, Lemma 1 yields a $\tau_0$ such that $\|\tau_0 \cdot g_1' - \hat{c}\| \leq \epsilon/2$, which ensures

$$\|\tau \cdot g_1' - \hat{c}\| \|s_1 - \bar{c}\| \leq 2\|\tau \cdot g_1' - \hat{c}\| \leq \epsilon,$$

for all $\tau \geq \tau_0$. Hence $\bar{c}$ is at least an $\frac{\epsilon}{\tau}$-PNE for all $\tau \geq \tau_0$ (w.l.o.g. $\tau_0 \geq 1$).

The above can be used to obtain a bound on $\delta := \|s_1 - \bar{c}\|$ for $s_1 \in \Psi(\bar{c})$. Using orthogonality

$$\Delta = \langle (I - \bar{c}\bar{c}^\top)g_1', s_1 - \bar{c} \rangle + \langle \bar{c}, g_1' \rangle \langle \bar{c}, s_1 - \bar{c} \rangle$$
$$\leq \tau^{-1}\|s_1 - \bar{c}\| \left[ \|(I - \bar{c}\bar{c}^\top)\tau \cdot g_1'\| - \frac{1}{2}\langle \bar{c}, \tau \cdot g_1' \rangle \|s_1 - \bar{c}\| \right],$$

by the triangle inequality, and $\langle \bar{c}, s_1 - \bar{c} \rangle = \frac{1}{2}(2\langle \bar{c}, s_1 \rangle - 2) = -\frac{1}{2}\|s_1 - \bar{c}\|^2$ by $s_1, \bar{c} \in S^{d-1}$. The terms in the square bracket on the r.h.s. can be bounded using the Pythagoras' theorem

$$\|\tau \cdot g_1' - \hat{c}\|^2 = \|(I - \bar{c}\bar{c}^\top)(\tau \cdot g_1' - \hat{c})\|^2 + \|\bar{c}\bar{c}^\top(\tau \cdot g_1' - \hat{c})\|^2$$
$$= \|(I - \bar{c}\bar{c}^\top)\tau \cdot g_1'\|^2 + |\langle \bar{c}, \tau \cdot g_1' \rangle - \|\hat{c}\||^2$$

where we used $(I - \bar{c}\bar{c}^\top)\hat{c} = 0$ and $\|\bar{c}\| = 1$. Because $\|\tau \cdot g_1' - \hat{c}\| < \epsilon$, the same is true for (the square roots of) both terms on the r.h.s. above. By a simple algebraic manipulation of these inequalities

$$\tau \cdot \Delta < \delta \left[ \epsilon - \frac{\delta}{2}(\|\hat{c}\| - \epsilon) \right]. \tag{5}$$

The r.h.s. is positive only when $0 < \delta < 2\epsilon/(\|\hat{c}\| - \epsilon)$. Since $\epsilon$ in Equation (5) is only used as an upper bound on $\|\tau \cdot g_1' - \hat{c}\|$, and Lemma 1 tells us this norm converges to zero, $\delta \to 0$ as $\tau \to \infty$. $\qquad \square$

**Proposition 1.** *A PNE always exists in $n = d = 2$ non-negative* hardmax *games, but may not without non-negativity or when $d > 2$. For $n = 2$ non-negative* softmax *games with $\hat{c} := \frac{1}{n}(1 - \frac{1}{n})\mathbb{E}[c] \neq 0$, the only possible PNE is $s_1 = s_2 = \bar{c}$ with $\bar{c} := \hat{c}/\|\hat{c}\|$ (independently of $d$), but a PNE may not exist. When $n > 2$, non-negative softmax games can have a PNE other than $s_1 = \cdots = s_n = \bar{c}$.*

*Proof of Proposition 1.* **(I) Hardmax:** For **existence when** $n = d = 2$, let $\theta_c$ be the angle of $c$ from (w.l.o.g.) $[1, 0]$, and let $A \subset C$ denote the set of angles such that for every $\theta_m \in A$, $\mathbb{P}(\theta_c \leq \theta_m) \geq \frac{1}{2}$ and $\mathbb{P}(\theta_c \geq \theta_m) \geq \frac{1}{2}$, with $\mathbb{P}$ implied by the underlying $P_c$. Then any $(s_1, s_2) \in A \times A$ is a PNE.

For **non-existence when** $d > 2$, consider $d = 3$ and $P_c = \frac{1}{3}\sum_{j=1}^{3}\delta_{c_j}$ where $c_j$ are the three canonical basis vectors. Assume $s = (s_1, s_2)$ is a PNE. Disregards of $s_1$ location, there will be a point $s_2$ on the great circle connecting the two most distant points from $s_1$ (break ties arbitrarily) which is closer to both of the two. Hence $u_2(s) \geq 2/3$ by the assumption that $s$ is a PNE. The same argument implies $u_1(s) \geq 2/3$. This is a contradiction since $\sum_i u_i(s) = 1$ by definition.

For **non-existence without non-negativity** in $d = 2$, see the hardmax part of the Theorem 2 proof.

**(II) Softmax:** In the $n = 2$ **case**, a necessary condition for $s = (s_1, s_2)$ to be a PNE is that the Riemannian gradients of the utility, $(I - s_i s_i^\top)g_i$ with $g_i = \nabla_{s_i} u_i(s)$, are zero. Since $\nabla_{s_i} u_i(s) = \tau^{-1}\mathbb{E}[p_i(c)(1 - p_i(c))c]$, $g_i$ belongs to the first orthant by the definition of a non-negative game, and it is not zero (for $\tau > 0$, all probabilities lie in $(0, 1)$, and $c$ is *not* a.s. zero since we assumed $\mathbb{E}[c] \neq 0$). Hence the Riemannian gradients can only be zero if $s_i \propto g_i$, and in particular $s_i = g_i/\|g_i\|_2$ because this is the direction which makes dot products with all vectors in the first orthant positive.

Crucially, $g_1 = g_2$ in 2-player games due to the symmetry of $p_1(c)(1 - p_1(c)) = p_1(c)p_2(c) = p_2(c)(1 - p_2(c))$. Therefore at a PNE, $s_1 = s_2$ in which case $p_i(c) = \frac{1}{2}$ for all $c$. Thus $g_i(s) \propto \mathbb{E}[c]$, implying $s_1 = s_2 = \bar{c}$ is the only possible PNE. To show it may not be a PNE, consider $P_c = \frac{1}{3}(2\delta_{c_1} + \delta_{c_2})$ for arbitrary non-zero $c_1 \neq c_2$ in the first orthant. Then $\bar{c} \propto 2c_1 + c_2$ with $u_1(\bar{c}, \bar{c}) = u_2(\bar{c}, \bar{c}) = 1/2$. Fixing $s_1 = c_1/\|c_1\|_2$ and taking $\tau \downarrow 0$, we get $u_1(s_1, \bar{c}) \to 2/3$, which means there exists a $\tau > 0$ for which $s_1 = c_1/\|c_1\|_2$ is a strict improvement over $s_1 = \bar{c}$ when $s_2 = \bar{c}$.

For the $n > 2$ **case**, we focus on a two-dimensional $n = 4$ game with $P_c = \frac{1}{2}(\delta_{c_1} + \delta_{c_2})$ with $c_1 = [1, 0]^\top$ and and $c_2 = [0, 1]^\top$ (the two canonical basis vectors). In particular, we investigate existence of NE of the form $s_1 = s_2$ and $s_3 = s_4$. Since $d = 2$, the strategies are restricted to $S^1$, which means we can use polar coordinates to parameterize $s_i = \varphi(\theta_i) := [\cos(\theta_i), \sin(\theta_i)]^\top$. We will further restrict our attention to the symmetric case $\theta_1 = \theta_2 = \theta$ and $\theta_3 = \theta_4 = \frac{\pi}{2} - \theta$ for some $\theta \in [0, \frac{\pi}{4}] =: K$. This allows us to define

$$Q := \begin{pmatrix} 0 & -1 \\ 1 & 0 \end{pmatrix},$$

and look for values of $\theta \in K$ where (w.l.o.g.) $f(\theta) := \frac{\partial u_1(s)}{\partial s_1}\frac{\partial s_1}{\partial \theta_1}|_{\theta_1 = \theta} = \langle g_1, Qs_1 \rangle$ is equal zero. Note that in the definition of $f$, all $s_i$ and $g_i$ vary with $\theta$ according to the relationship $s_i = \varphi(\theta_i)$ with $\theta_1 = \theta_2 = \theta$ and $\theta_3 = \theta_4 = \frac{\pi}{2} - \theta$. However, $f(\theta)$ is only the derivative of $u_1(s)$ w.r.t. $\theta_1$, ignoring the dependence of $s_2$, $s_3$ and $s_4$ on $\theta$. This definition of $f$ means that only the roots of $f$ can possibly be NE. The next lemma will help us locate these roots.

**Lemma 2.** *For a sufficiently small $\tau > 0$, $f: \theta \mapsto \langle g_1, Qs_1 \rangle$ is strictly* convex *on $K$.*

*Proof of Lemma 2.* It is sufficient to prove that $f'' > 0$ on $K$. For this, observe $f'(\theta) = \|Qs_1\|^2_{H_1} - \langle g_1, s_1 \rangle$ where $H_1 := \nabla^2_{s_1} u_1(s)$, and $f''(\theta) = \|Qs_1\|^2_{\nabla_{\theta_1} H_1} - \langle Qs_1, 3H_1 s_1 + g_1 \rangle \geq \|Qs_1\|^2_{\nabla_{\theta_1} H_1} - 3\|H_1\|_2 - \|g_1\|_2$, where by construction

$$g_1 = \frac{1}{2\tau}\begin{pmatrix} p_1(c_1)(1 - p_1(c_1)) \\ p_1(c_2)(1 - p_1(c_2)) \end{pmatrix},$$

$$H_1 = \frac{1}{2\tau^2}\begin{pmatrix} (1 - 2p_1(c_1))p_1(c_1)(1 - p_1(c_1)) & 0 \\ 0 & (1 - 2p_1(c_2))p_1(c_2)(1 - p_1(c_2)) \end{pmatrix}$$

$$\nabla_{\theta_1} H_1 = \frac{1}{2\tau^3}\begin{pmatrix} (1 - 6p_1(c_1)(1 - p_1(c_1)))p_1(c_1)(1 - p_1(c_1)) & 0 \\ 0 & (1 - 6p_1(c_2)(1 - p_1(c_2)))p_1(c_2)(1 - p_1(c_2)) \end{pmatrix},$$

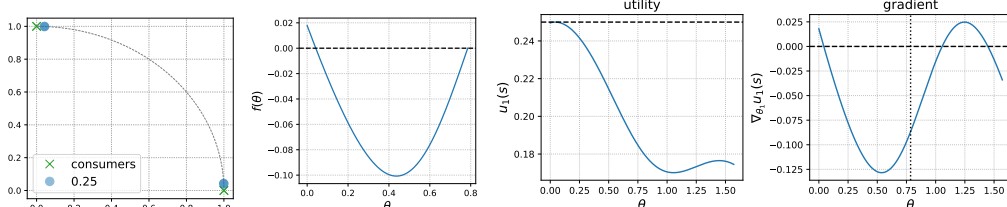

Figure 7: $n > 2$ **softmax case from the proof of Proposition 1. Left:** Symmetric PNE location (here for $\tau = \frac{1}{4}$). **Middle left:** Plot of $f(\theta) = \langle g_1(s), Qs_1 \rangle$ with $s_1 = s_2 = \varphi(\theta)$ and $s_3 = s_4 = \varphi(\frac{\pi}{2} - \theta)$. **Right:** Plot of utility and its gradient for all possible defection strategies $s_1 = \varphi(\theta_1)$ with $s_2, s_3, s_4$ kept put in the positions defined by $\theta_\tau^\star$ from the left plots. Vertical line shows $\frac{\pi}{4}$ (right end of $K$).

Hence $\|Qs_1\|_{\nabla_{\theta_1} H_1}^2 \sim \tau^{-3}$, $\|H_1\|_2 \sim \tau^{-2}$, and $\|g_1\|_2 \sim \tau^{-1}$, implying that for $\tau$ low enough, the positive term $\|Qs_1\|_{\nabla_{\theta_1} H_1}^2$ dominates (using that all expressions share the term $p_1(c)(1 - p_1(c))$), and thus after dividing and observing $\tau \to 0$ gives $p_1(c)$ close to either one or zero, we get that all the terms scale as $p_1(c)(1 - p_1(c))/\tau^k$ for the appropriate $k \in \{1, 2, 3\}$). $\qquad\square$

Lemma 2 implies there are at most two NE ($f$ is strictly convex, and $f(\theta) = 0$ is a necessary condition). At $\theta = \frac{\pi}{4}$, $s_1 = s_2 = s_3 = s_4 = \bar{c}$ by definition, which we know is a critical point, so $f(\frac{\pi}{4}) = 0$. Since $g_1 \propto \mathbb{E}[p_1(c)(1 - p_1(c))c] \neq 0$ for any $\tau > 0$ ($c$ cannot be a.s. 0 by $\mathbb{E}[c] \neq 0$ and the non-negativity assumption), $f(0) = \langle g_1, Qs_1 \rangle > 0$ ($s_1 = \varphi(0) = e_1 = [1, 0]^\top$). The other possible root of $f$ thus could only be in the interior $(0, \frac{\pi}{4})$ of $K$. For small enough $\tau$, moving from $\theta = \frac{\pi}{8}$ towards $e_1 = [1, 0]^\top$ will increase utility, implying $f(\frac{\pi}{8}) < 0$. Hence there exists $\tau > 0$ and $\theta_\tau^\star \in (0, \frac{\pi}{8})$ s.t. $f(\theta_\tau^\star) = 0$ by the mean value theorem.

So far we have established that $s_1 = s_2 = \varphi(\theta_\tau^\star)$, $s_3 = s_4 = \varphi(\frac{\pi}{2} - \theta_\tau^\star)$ is a *local* NE for the corresponding small $\tau$. By symmetry, it is sufficient to check if there is a defection strategy for $s_1$. Any defection to $\theta_1 \in (\theta_\tau^\star, \frac{\pi}{2} - \theta_\tau^\star]$ will result in $p_1(c) < \frac{1}{4}$ for both $c = c_1, c_2$, and thus worse utility. Defection to $(\frac{\pi}{2} - \theta_\tau^\star, \frac{\pi}{2}]$ will not yield utility greater than defection to $[0, \theta_\tau^\star)$ since $s_3 = s_4 = \varphi(\frac{\pi}{2} - \theta_\tau^\star)$, so it is sufficient to focus on $\theta_1 \in [0, \theta_\tau^\star)$. Here

$$
\begin{aligned}
\nabla_{\theta_1} u_1(s) &= \langle g_1, Qs_1 \rangle \\
&\propto p_1(c_2)(1 - p_1(c_2)) \cos(\theta_1) - p_1(c_1)(1 - p_1(c_1)) \sin(\theta_1) \\
&\geq p_1(c_1)(1 - p_1(c_1))[\cos(\theta_1) - \sin(\theta_1)],
\end{aligned}
$$

since $p_1(c_1)$ grows quicker than $p_1(c_2)$ decays. By construction, $\theta_\tau^\star < \frac{\pi}{4}$, and we know $\cos(\theta) - \sin(\theta) > 0$ for $\theta \in [0, \frac{\pi}{4})$. In other words, the utility of $s_1$ is strictly increasing on $\theta_1 \in [0, \theta_\tau^\star)$, i.e., none of the corresponding $s_1 = \varphi(\theta_1)$ is an improvement. Hence $s_1 = s_2 = \varphi(\theta_\tau^\star)$, $s_3 = s_4 = \varphi(\frac{\pi}{2} - \theta_\tau^\star)$ is a NE. (The construction is illustrated in Figure 7.) $\qquad\square$

**Proposition 2.** *Every $\tau > 0$ exposure game with $\mathbb{E}[c] \neq 0$ has a critical point at $s_1 = \ldots = s_n = \bar{c}$.*

*Proof of Proposition 2.* When $s_1 = \cdots = s_n = \bar{c}$, the gradient from Equation (4) is the same for all producers, and it is proportional to $(I - \bar{c}\bar{c}^\top)\hat{c}$. This is equal to zero by $\bar{c} = \hat{c}/\|\hat{c}\|$. $\qquad\square$

## A.2 Inline statements

**Lemma 3.** *The distribution from the part (I) of the proof of Theorem 2—$d = 2$, $P_c = \frac{1}{3}\sum_{j=1}^3 \delta_{c_j}$ with $c_j \neq c_k$ $\frac{2\pi}{3}$ apart—admits a mixed NE $P_1 = P_2 = P_c$ at $\tau = 0$.*

*Proof of Lemma 3.* By symmetry, $u_i(P_c, P_c) = \frac{1}{2}$, $\forall i$. Since for any $s_1 \in \text{supp}(P_c) = \{c_1, c_2, c_3\}$

$$
\mathbb{E}_{c, s_2}[u_1(s_1, s_2) \,|\, s_1] = \frac{1}{3}[u_1(s_1, c_1) + u_1(s_1, c_2) + u_1(s_1, c_3)] = \frac{1}{3}[1 \cdot \frac{1}{2} + 2 \cdot \frac{1}{3}(1 + \frac{1}{2} + 0)] = \frac{1}{2},
$$

all we need is to show that $\mathbb{E}_{c,s_2}[u_1(s_1,s_2) \mid s_1] \leq \frac{1}{2}$ for any $s_1 \notin \text{supp}(P_c)$. W.l.o.g. assume $s_1$ lies on the geodesic connecting $c_1$ and $c_3$ (i.e., on the arc opposite of $c_2$). Such an $s_1$ is closer to $c_1$ and $c_3$ than $c_2$ ($u_1(s_1,c_2) = \frac{2}{3}$), but is further from $c_1$ and $c_2$ (resp. $c_3$ and $c_2$) than $c_3$ (resp. $c_1$). Hence

$$\mathbb{E}_{c,s_2}[u_1(s_1,s_2) \mid s_1] = \tfrac{1}{3}[u_1(s_1,c_1) + u_1(s_1,c_2) + u_1(s_1,c_3)] = \tfrac{1}{3}[1 \cdot \tfrac{2}{3} + 2 \cdot \tfrac{1}{3}] = \tfrac{4}{9}.$$

Since $\frac{4}{9} < \frac{1}{2}$, $s_1$ has no incentive to move any of its mass away from $\text{supp}(P_c)$. □

## B   HARDMAX GAMES

In this section we present two different algorithms for finding mixed Nash equilibria in two-player hardmax games. We note that the set of allowable mixed strategies must be restricted in some way since certain distributions with support on the unit-sphere $S^{d-1}$ require infinite storage. Hence, our first algorithm finds a mixed NE for a discretized strategy space, while our second algorithm considers settings where $P_c$ is discrete and finds a mixed NE with support over a finite number of pure strategies in the original non-discretized space, assuming such a mixed NE exists.

We caution that both of these algorithms can only find mixed NE for small exposure games due to their poor scaling properties. We list them here to highlight the difficulty of solving hardmax games when compared to the softmax setting and to serve as inspiration for future research into more efficient algorithms.

### B.1   DISCRETIZED GAMES

We first consider the setting where both players may only choose mixed strategies with support over a finite subset $A = \{s^{(1)}, s^{(2)}, \ldots, s^{(m)}\} \subset S^{d-1}$ of pure strategies. This setting includes embeddings that are represented using floating point numbers although $A$ will be very large. In this case the mixed strategy of the players can be expressed as an $m$-dimensional probability vector $\pi_i$ with $\pi_{ij} = P_i(s^{(j)})$. Since there are a finite set of pure strategies a mixed NE is guaranteed to exist (Nash Jr, 1950). Furthermore since this is a two-player constant-sum game we can find a mixed NE by solving the following linear program (Dorfman, 1951)

$$
\begin{aligned}
\underset{\alpha}{\text{maximize}} \quad & \alpha \\
\text{subject to} \quad & Ux \geq \alpha\mathbf{1} \\
& \mathbf{1}^\top x = 1 \\
& x_i \geq 0, \ i = 1, \ldots, m,
\end{aligned}
$$

where $U_{ij} = u_1(s^{(i)}, s^{(j)})$. The strategies where $\pi_1 = \pi_2 = x$ correspond to a mixed NE. While such a problem is simple to formulate and solve, the number of possible strategies grows rapidly with $d$ for most discretization schemes. For example, we might create a uniform grid of $k$ points over each spherical coordinate, in which case we will have $m = k^{d-1}$ pure strategies to consider.

### B.2   FINITE SUPPORT

Next, we consider the setting where both players choose mixed strategies with support over at most $m$ pure strategies, and the support of $P_c$ is over $l$ points, $\text{supp}(P_c) = \{c_1, c_2, \ldots, c_l\}$. Unlike in the discretized case, the players may choose any pure strategy that lies on $S^{d-1}$. We begin by outlining a method that, given a mixed strategy $P$, finds all pure strategies $D$ that dominate it: $\mathbb{E}_{s_1 \sim P}[u_1(s_1, s_2) - u_2(s_1, s_2)] < 0$ for all $s_2 \in D$. By symmetry, we w.l.o.g. assume Player 1 provides the mixed strategy. We will then use this method as a subroutine to find a mixed NE.

**Lemma 4.** $(P_1, P_2)$ is a mixed NE if and only if $\mathbb{E}_{s_1 \sim P_1}[u_1(s_1, s)] \geq \frac{1}{2}$ and $\mathbb{E}_{s_2 \sim P_2}[u_2(s, s_2)] \geq \frac{1}{2}$ for all pure strategies $s \in S^{d-1}$.

*Proof of Lemma 4.* Assume $(P_1, P_2)$ is a mixed NE, then by definition $\mathbb{E}_{(s_1,s_2)\sim(P_1,P)}[u_1(s_1,s_2)] \geq \frac{1}{2}$ for all mixed strategies $P \in \mathcal{P}(S^{d-1})$, since each pure strategy is also a mixed strategy it follows that $\mathbb{E}_{s_1 \sim (P_1, s)}[u_1(s_1, s)] \geq \frac{1}{2}$ for all $s \in S^{d-1}$. Similarly for Player 2.

Now assume we have two mixed strategies $(P_1, P_2)$ such that $\mathbb{E}_{s_1 \sim P_1}[u_1(s_1, s)] \geq \frac{1}{2}$ and $\mathbb{E}_{s_2 \sim P_2}[u_2(s, s_2)] \geq \frac{1}{2}$ for all $s \in S^{d-1}$. Given a mixed strategy $P \in \mathcal{P}(S^{d-1})$ it follows that

$$\mathbb{E}_{(s_1, s_2) \sim (P_1, P)}[u_1(s_1, s_2)] = \mathbb{E}_{s_2 \sim P}[\mathbb{E}_{s_1 \sim P_1}[u_1(s_1, s_2)]]$$

$$= \int_{S^{d-1}} \mathbb{E}_{s_1 \sim P_1}[u_1(s_1, s_2)] dP(s_2)$$

$$\geq \int_{S^{d-1}} \frac{1}{2} dP(s_2) = \frac{1}{2}.$$

Similarly for Player 2. $\qquad\qquad\qquad\qquad\qquad\qquad\qquad\qquad\qquad\qquad\qquad\qquad\qquad\qquad$ $\square$

Lemma 4 allows us to only consider pure strategies when checking if strategies are mixed NE.

Now given a mixed strategy $P$ with finite support $\mathrm{supp}(P) = \{s^{(1)}, s^{(2)}, \ldots, s^{(m)}\}$ we can find every subset of $S^{d-1}$ that does not satisfy the condition in Lemma 4. By noting that any arbitrary strategy $s$ can be either closer, farther, or at the same distance from a consumer as a given $s^{(i)}$; we see that each $s^{(i)}$ partitions $S^{d-1}$ into $3^l$ disjoint partitions based upon the distance of the strategies to each consumer $c_k$. That is, $\mathcal{X}^{(i)} = \{X_1^{(i)}, X_2^{(i)}, \ldots, X_{3^l}^{(i)}\}$, with $X_j^{(i)}$ satisfying

$$j_k = \begin{cases} 2 & \texttt{if} \ \langle s^{(i)}, c_k \rangle > \langle s, c_k \rangle \\ 1 & \texttt{if} \ \langle s^{(i)}, c_k \rangle = \langle s, c_k \rangle \\ 0 & \texttt{if} \ \langle s^{(i)}, c_k \rangle < \langle s, c_k \rangle, \end{cases}$$

for all pure strategies $s \in X_j^{(i)}$, where $j_k$ is the $k$-th digit in the ternary representation of $j$. By considering all $m$ partitions created by the strategies in $\mathrm{supp}(P)$, we can further partition the space into $3^{lm}$ disjoint partitions $\mathcal{Y} = \{Y_1, Y_2, \ldots, Y_{3^{lm}}\}$ with $Y_i = \bigcap_{j=1}^{m} X_{i_j}^{(j)}$ where $i_j$ is the $j$-th digit of the $3^l$-ary representation of $i$.

For every $Y \in \mathcal{Y}$ we have $\mathbb{E}_{s_1 \sim P}[u_1(s_1, s)] = \mathbb{E}_{s_1 \sim P}[u_1(s_1, s')]$ for all $s, s' \in Y$ by construction. Thus, we can find the set of all pure strategies $D$ that dominate $P$ by iterating over $\mathcal{Y}$, testing a single point in each partition, and taking unions:

$$Z = \left\{ Y \in \mathcal{Y} : s \in Y \implies \mathbb{E}_{s_1 \sim P}[u_1(s_1, s)] < \frac{1}{2} \right\}, \quad D = \bigcup_{Y \in Z} Y.$$

It follows from Lemma 4 that $(P, P)$ is a mixed NE if and only if $D$ is empty.

Finally, we outline a method to find mixed NE. We first note that for every positive integer $m$, every pure strategy $s \in S^{d-1}$ defines a feasible set $F_s$ of all mixed strategies with support over at most $m$ pure strategies that are not dominated by $s$, that is:

$$F_s = \left\{ P = \sum_{i=1}^{m} \pi_i \delta_{s^{(i)}} : \sum_{i=1}^{m} \pi_i u_1(s^{(i)}, s) \geq \frac{1}{2} \right\},$$

where $\pi$ is an $m$-dimensional probability vector. It follows from Lemma 4 that if $P$ is mixed strategy with support over at most $m$ points then $(P, P)$ is a mixed NE if and only if $P \in \bigcap_{s \in S^{d-1}} F_s$. We can frame finding such a strategy $P$ as an optimization problem

$$\underset{\mathbf{P} \subset \mathcal{P}}{\text{minimize}} \quad |\mathbf{P}|$$

$$\text{subject to} \quad \mathbf{P} \cap F_s \neq \emptyset, \ s \in S^{d-1},$$

where $\mathcal{P}$ is the set of all mixed strategies with support over at most $m$ pure strategies. An optimal solution with more than one element in $\mathbf{P}$ indicates that there does not exist a mixed strategy with support over $m$ points or fewer, whereas if $|\mathbf{P}| = 1$ then $(P, P)$ is a mixed strategy where $P$ is the singleton element in $\mathbf{P}$.

This is an instance of the *implicit hitting set problem*. Hence, we can use the algorithm proposed in Section 2.1 of Chandrasekaran et al. (2011) to solve the above optimization problem. Their

algorithm assumes an oracle that, given a proposed subset $\mathbf{P} \subseteq \mathcal{P}$ will return a subset $F_s$ that is not hit $\mathbf{P} \cap F_s = \emptyset$ or will certify $\mathbf{P}$ as a feasible solution to the above optimization problem. We can easily achieve this by finding all dominating pure NE using our proposed method above for each $P \in \mathbf{P}$ and taking the intersection of the resulting sets. If the intersection is empty then $\mathbf{P}$ is a feasible solution, otherwise every element in the intersection represents a subset $F_s$ that has not been hit by $\mathbf{P}$.

# C  EXPERIMENTS

## C.1  SETUP

The LastFM dataset was preprocessed by Shakespeare et al. (2020). Original larger scale sweep was executed with $n \in \{10, 25, 100, 500, 1500\}$, $d \in \{3, 10, 50, 100\}$, stepsize in $\{10^{-3}, 10^{-2}, 10^{-1}\}$, and $\tau \in \{10^{-2}, 10^{-1}, 0.25, 0.5, 1.0\}$. We only used 2 random seeds for the recommender, and 3 random seeds for our LNE-finding algorithm (i.e., 6 runs in total per configuration). For the reported results, stepsize sweep was restricted to $\{10^{-2}, 10^{-1}\}$; the number of steps was upper bounded by 50,000 (all runs have successfully converged to a fixed point as mentioned). While our code contains an option to `scale_lr_by_temperature` (see the `config.py` file in the provided code), which multiplies the stepsize by $\tau$ before its use, we did not use this option in the experiments.

The second-order Riemannian test (Definition 3) is implemented in `manifold.py`. Defining the tangent space projection $\Pi_i := (I - s_i s_i^\top)$, we consider a candidate strategy profile $s \in (S^{d-1})^n$ as *violating* the second order test if any of the Riemannian gradients $\Pi_i \nabla_{s_i} u_i(s)$ had $\ell^2$-norm higher than $10^{-5} \cdot \sqrt{d}$, *or* the Riemannian Hessian $\Pi_i [\nabla_{s_i}^2 u_i(s)] \Pi_i - \langle s_i, \nabla_{s_i} u_i(s) \rangle \Pi_i$ had a strictly positive eigenvalue (no tolerance used here).

The final MovieLens and LastFM experiments were run on 72 AWS machines, each with 4 CPU cores, for 5 hours. Including preliminary and failed runs, we used over 50K CPU hours.

## C.2  OPTIMIZER

The gradient ascent optimization technique (Singh et al., 2000; Balduzzi et al., 2018) we employ is very similar to standard gradient descent algorithm from machine learning literature. Here we provide a short description of the similarities and differences between the two.

The optimizer we use *simultaneously* runs $n$ independent gradient descent optimizers, each following the gradient of the utility $u_i(s)$ w.r.t. $\theta_i$, $i \in [n]$, as described around Equation (4) (recall $s_i = \theta_i / \|\theta_i\|$). $\theta_{i,t+1}$ is obtained using $\theta_{j,t}$ for all $j \neq i$, i.e., the locations of the other producers from the last step. All $n$ optimizers execute these steps at the same time, iterating until *all* of them converge. See `optimisation.py`, particularly the `optax_minimisation` method, for more details.

## C.3  ADDITIONAL PLOTS

Appendix C.3.1 contains plots where the second-order test confirmed and LNE. Appendix C.3.2 then offers comparison to a third ranking algorithm: standard matrix factorization (MF; Koren et al., 2009), i.e., PMF with additional bias terms. The bias terms effect interpretation of $\tau$ values, and we also ignore them when running the LNE-finding algorithm. This makes the comparison with PMF and NMF difficult, which is why we excluded MF from the main text. Results in Appendix C.3.2 again contain runs where the second-order test did not rule out a LNE.

### C.3.1  LNE CONFIRMED BY THE SECOND-ORDER TEST

As mentioned, the plots shown in the main body of the paper are for runs where the second-order Riemannian test did not rule out that the found pure strategy profile is a LNE. Here we show exactly the same plots with only the runs where the test confirmed a LNE. The difference is that here we exclude the runs where the Riemannian Hessian had at least one zero eigenvalue associated with a direction perpendicular to $s_i$, for at least one $i \in [n]$. As you see below, this had little effect on the LastFM results, but has non-negligibly reduced the number of admitted runs for MovieLens.

### C.3.2  MATRIX FACTORIZATION (PMF WITH BIASES) RESULTS

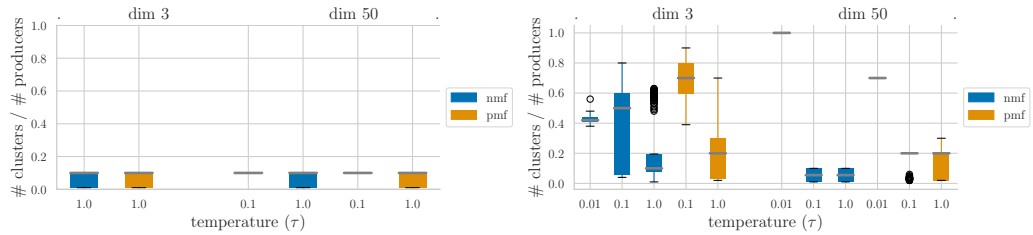

Figure 8: A counterpart to Figure 4 with runs where LNE test was inconclusive excluded.

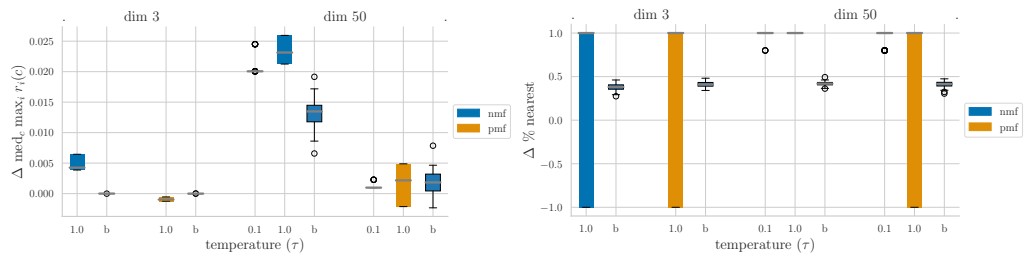

Figure 9: A counterpart to Figure 5 with runs where LNE test was inconclusive excluded.

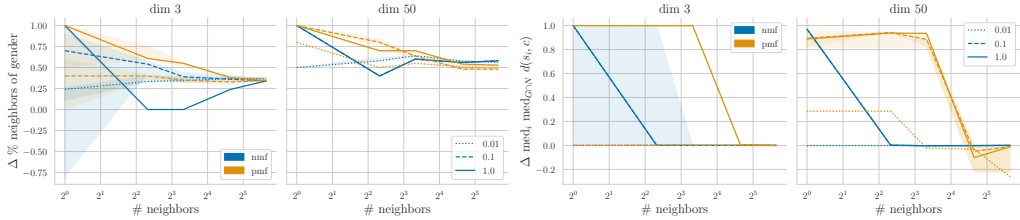

Figure 10: A counterpart to Figure 6 with runs where LNE test was inconclusive excluded.

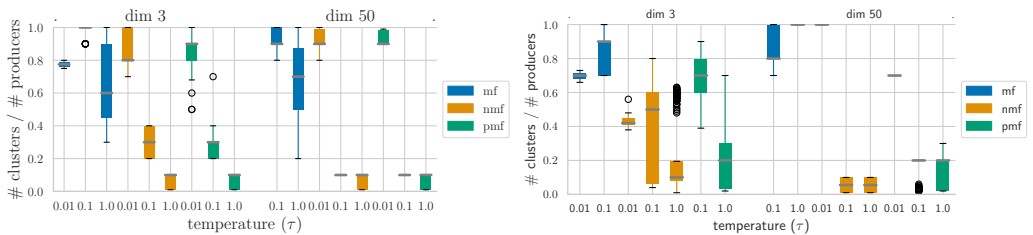

Figure 11: A counterpart to Figure 4 with added MF results.

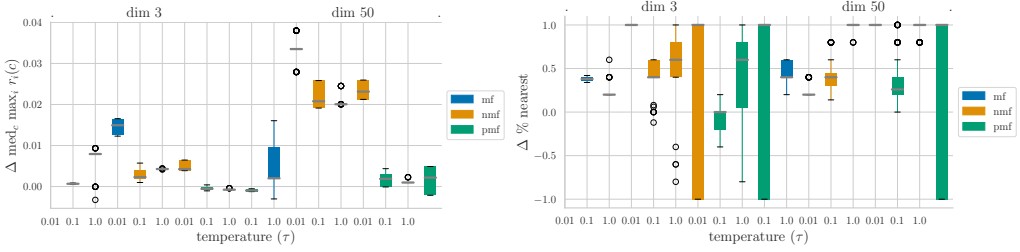

Figure 12: A counterpart to Figure 5 with added MF results. Baselines omitted to reduce clutter.

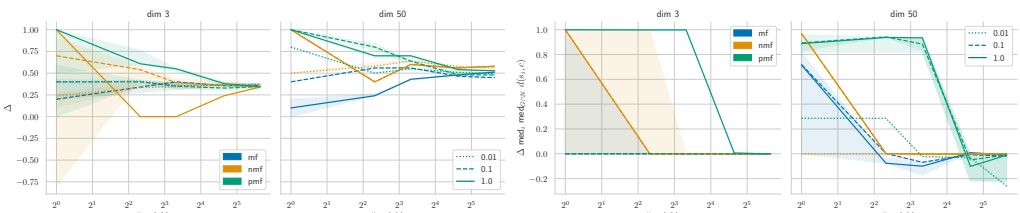

Figure 13: A counterpart to Figure 6 with added MF results.

