# OpenReview forum: "Modeling content creator incentives on algorithm-curated platforms"
_ICLR.cc/2023/Conference — ICLR 2023 notable top 5%_

### Official Review · Reviewer_49Ro · 2022-10-24

**Confidence:** 4
**Correctness:** 4
**Technical Novelty And Significance:** 4
**Empirical Novelty And Significance:** 3
**Recommendation:** 10

**Clarity, Quality, Novelty And Reproducibility:**

### Clarity

- Page 2: “We find the incentivized content exhibits a strong dependence between algorithmic exploration and content diversity (confirming our theory), and between model expressivity and bias towards gender-based user and creator groups.” This sentence is not very clear when reading it but it is detailed later in the paper and only then became clear to me.
- Page 4, right after Theorem 1: "via a classic result by Glicksberg (1952)". Could you please point more precisely or detail it in the appendix?
- Figure 3, plot C: how does this plot demonstrates lack of quasi-concavity? I might be missing something but I think I disagree.
- Page 5: "Figure 3A illustrates the non-existence result." Sure but it is not trivial and only detailed later in the appendix.
- Page 8: “Since the effect of $τ$ on rating models varies, we also include baseline values (labeled by ‘b’) computed using the original learned item embeddings (i.e., item locations before strategic adaptation).” Could you rephrase it as it doesn't appear clear to me.
- Page 15: In the Appendix, in the proof of Lemma 1: “By monotonicity, $(s_1^⋆, c^⋆)$ will be a maximizer $∀τ^′ ≥ τ$ .” I am not sure I understand. Could you be more explicit?
- Page 15, before equation $(5)$: The equality $|| \tau \cdot g_1' - \hat{c}||^2 = || (I - \bar{c}\bar{c}^{\top}) \tau \cdot g_1' || + |\left< |\bar{c}, \tau \cdot g_1' \right> - ||\hat{c}|| |^2$ is unclear.
- Too many undefined abreviations: "w.l.o.g.", "a.s.". Did I miss their definition?
- Page 17, at the very top: "since $p_1(c_1)$ grows quicker than $p_1(c_2)$ decays." Could you please be more explicit?

### Quality

This work gives an impressive number of highly relevant references, very clever proofs, and the authors regularly link their work to real case usage throughout the paper. Although some arguments are given too quickly, this work is of rare quality.

### Novelty

This paper generalizes the context of previous work and complement concurrent ones. The results obtained appear novel to the best of my knowledge.

### Reproducibility

The code is given but no README file nor environment file. The code seems well written which is a hint towards reproducibility.

**Strength And Weaknesses:**

### Strengths

- The proofs of this paper are very (sometimes too) elegant.
- The model proposed would be highly beneficial when used to assess recommendation systems and potential bias that could emerge prior to deploying them to production.
- The paper is well written and, except for the proofs and a few other points, easy to follow.
- As the author claim: their work “apply equally to classical matrix factorization and deep learning-based systems.” Which is appreciable, given how deep learning-based systems are still underresearched.
- The authors acknowledge the limitations of their model and do not present it as a miracle solution.

### Weaknesses

- Page 2: You define $S^{d-1}$ after using it. Please define it before/within Definition 1.
- Figure 3, plot B & C: if 1 producer is at midpoint, shouldn't the utility reach $\frac{1}{n}$ at $\lambda = 0.5$? Or did you not normalize the vector $s_1$?
- Page 6: You mention “Riemannian second derivative test” Can you please be more specific? The reference is long, dense, and do not mention this term.
- Page 17, Figure 7: It seems to me like there is an issue with the two central plots as $f(\theta) = \nabla_{\theta_1} u_1 (s)$ but the plot do not seem to show this relationship. $f(\frac{\pi}{4}) = 0$ but $u_1(s)$ appears decreasing at that value of $\theta = \frac{\pi}{4}$.
- Page 18: It seems to me like the partitions $\mathcal{X}^{(i)}$ and $\mathcal{Y}$ are unnecessarily complex, I didn't fully get the grasp on them. Isn't there a simpler way to deal with it?

Not really weaknesses:

- Page 3: "motivated creators will actively optimize their exposure using trial-and-error, making complete information and full-control an imperfect yet not unreasonable model of their behavior." Or is it? This trial-and-error is done in an always-changing environment (many hours of video uploaded every second on youtube), so is it that informative?
- Page 6: “Exploration effects are typically thought of as having negative impact on user experience through immediate reduction in quality of service as a result of suboptimal recommendations.” Being trapped in an "algorithmic bubble" with always the same type of recommendation is critisized and not appreciated.
- Proposition 1: isn't there a typo? "For $n=2$ *non-negative* softmax *games*". Shouldn't it be $n \geq 2$?
- Page 15, proof of proposition 1: $\mathbb{P}(\theta \geq \theta_m) = \frac{1}{2}$. What is $\theta$?
- Page 16, typo: $P_c = \frac{1}{2}(\delta_{c_1} + \delta c_2)$, should be $\delta_{c_2}$.
- Page 17, typo: "$c_3$ then $c_2$" should have been "than".
- Page 17, typo: "$c_1$ (resp. $c_3$)" it seems to me like they should be switched.



**Summary Of The Paper:**

This paper study how technical changes in recommendation algorithms - giving exposure of a group of producers' products to a group of users - can inventive strategic content creators. They model such context through an exposure game with strategic rational omniscient agents fully capable of changing their strategies. They demonstrate both theoretically and experimentally the impact of a parameter of their model on the Nash equilibrium to which the content creators can converge. The authors propose to use this game as an audit tool to determine the incentives of strategic creators before modifying a recommendation algorithm in production.

**Summary Of The Review:**

This paper is very pleasing to read. Although some proofs are highly challenging, this paper seems novel, gives many highly relevant references for the topic and the goal of the paper will be of great use as a tool to evaluate incentives towards content creators in recommendation models, which has many social implications. The quality of this paper excuses for the few unclear moments. I would be more than glad to raise my score once my concerns are raised.

EDIT: The authors addresses my concerns on their work, I increased my score in response.

---

> ### Author Response · Authors · 2022-11-19
> **Rebuttal part I**
>
> Thank you for taking the time to read and review our paper!
>
> Please see our responses below; we have started working on the corresponding changes (see uploaded revised version), and will incorporate all of them in the final revision & credit anonymous reviewers in the acknowledgements. If you have any further concerns, we would be keen to address them as well.
>
> ### General comments
>
> * **Re Figure 3, plot B & C**: Yes, the utility is for the _unnormalised_ vectors, which was only mentioned in the main text but not the caption. We are not normalising because plots B & C illustrate why standard results guaranteeing existence of PNE do not apply. In particular, these results require convexity of the strategy space, and quasi-concavity of the utility. Hence we are not normalising to show that even if our strategy space _was_ convex (i.e., a d-dimensional ball instead of a sphere), the quasi-concavity still does not hold. We will clarify this in the paper, thanks!
> * **Re Riemannian second derivative test on p.6**: Our reference was indeed imprecise. The relevant result is in lemma 5.41 in (Boumal, 2022), and the text below it. For completeness, more on the Riemannian gradient (proposition 3.61) and Reimannian Hessian (corollary 5.16) can also be found in (Boumal, 2022).
> * **Re Page 17, Figure 7**: This is an understandable misunderstanding of our notation. $f(\theta)$ gives $\frac{\partial s_1}{\partial \theta_1}^\top \nabla_{s_1} u_1(s)$ **when** $s = [\varphi(\theta), \varphi(\theta), \varphi(\frac{\pi}{2} - \theta), \varphi(\frac{\pi}{2} - \theta)]^\top$, i.e., as we change $\theta$, the positions of **all** producers change. This is useful, because it allows us to find candidate NE of the symmetric form we assumed in the proof. In contrast, the two plots on the right give $u_1(s)$ and $\frac{\partial s_1}{\partial \theta_1}^\top \nabla_{s_1} u_1(s)$ when $s_1 = \varphi(\theta)$ **but** $s_2 = \varphi(\theta_\tau^\star)$ and $s_3 = s_4 = \varphi(\frac{\pi}{2} - \theta_\tau^\star)$, i.e., all producers but $s_1$ are fixed. The gradient plot you're looking for is thus on the right, not the middle left. We will use a clearer notation and language in the next revision.
> * **Re Page 18 partitions X(i) and Y are unnecessarily complex**: We agree that the current definitions are not the easiest to grasp. We added some extra descriptive text which aims to address what these sets are meant to conceptually represent. However, we would be open to additional ways in which we could improve the clarity.
>
> * **Re page 3 exposure motivation**: We believe exposure maximisation is an imperfect but sensible place to start investigation (p. 3), since gaining exposure is necessary to realise both monetary and non-monetary goals. It is also the most common utility function used in prior literature (Section 1.2). Whether or not creators can—at least in the long term—reach local optima, is a great question and in our opinion an important direction for future research.
> * **Re page 6 exploration effects**: The relationship between exploration and user experience is indeed nuanced, which we allude to but don’t explore in detail due to the limited space. We will update the language to reflect your comment.
> * **Re Proposition 1 typo**: In proposition 2, we make statements separately for the $n=2$ case (in the sentence you point out) and the $n>2$ case (towards the end of the Proposition), so it is not a typo. Please let us know if you meant something different.
> * **Re page 15 $\theta$**: This is a typo, should have read $\theta_c$ instead.
> * **Re other typos**: Fixed, thank you!

---

> > ### Author Response · Authors · 2022-11-19
> > **Rebuttal part II**
> >
> > ### Clarity
> >
> > * **Re page 2 result description**: We rephrased to _“We find a strong dependence between algorithmic choices like embedding dimension and level of exploration, and properties of the incetivized content such as diversity (confirming our theory), and targeting of gender-based user and creator groups.”_ However, we are open to further suggestions.
> > * **Re page 4 Glicksberg ref**: Added pointer to section 2 in the cited Glicksberg (1952) paper. The cited result applies since $S^{d-1}$ is indeed a compact Hausdorff pure strategy space, and the utilities $u_i$ ($M_i$ in the notation of the cited paper) are continuous in $s$.
> > * **Re Figure 3, plot C**: Thanks for spotting this!  At the last minute, we’ve changed the figure from showing absolute values to showing the difference, but we accidentally computed it the wrong way. The figure is now changed, clearly showing the dip below the two endpoints, violating the quasi-concavity.
> > * **Re page 5 non-existence result**: We’ve added a pointer to the proof.
> > * **Re page 8 baseline**: We rephrased to “To help disentangle effect of exposure maximization, we also include statistics based on the original item locations (labeled by ‘b’), i.e., the content created before producers adapt to the recommender.”_  However, we are open to further suggestions.
> > * **Re page 15 monotonicity**: Thanks, we simplified the proof. Essentially, $c$ and $s_1’$ only affect the value through their dot product scaled by $\tau^{-1}$. As $\tau$ gets large, the dot product uniformly vanishes because both $s_1’$ and $c$ live on bounded domains. Please see the updated manuscript for details.
> > * **Re page 15 before equation (5)**: Thanks, we added more detail. Essentially, the equality holds by the parallelogram law/Pythagoras’ theorem, since we decompose the original vector into two orthogonal components, one parallel to $\bar{c}$ and the other its orthogonal complement.
> > * **Re undefined abbreviations**: Fixed (beginning of Appendix), thanks!

---

### Official Review · Reviewer_rzZd · 2022-11-03

**Confidence:** 4
**Correctness:** 3
**Technical Novelty And Significance:** 4
**Empirical Novelty And Significance:** 4
**Recommendation:** 8

**Clarity, Quality, Novelty And Reproducibility:**

The paper could do with another couple of rounds of revision to make it fully clear, and to remove some of the problems.
The experimental results are not explained in anywhere near enough detail to be reproducible.

**Strength And Weaknesses:**

Strengths:
- The topic is very interesting, and important.
- The authors use serious game-theoretical approaches to assess the likely outcome of the strategic considerations by content creators
- The results are, so far as I can tell on an extremely quick read, more or less correct.
- The results tell us something interesting - more explorative recommender systems can help to avoid over-targeting by producers, but might reduce the diversity of content that is created.

Weaknesses:
- There are several problems in the proofs (see below).
- Insufficient details are given about the experimental results. Section 3.2 was extremely difficult to follow - the plots are very small, with poorly explained axes; which concept of clustering is being used; etc etc.
- The claims about switching from non-negative to unconstrained feature spaces having a strong effect is only actually present in Proposition 1, in the case of two players and a two-dimensional feature space (since features are constrained to have norm 1, what has actually happened is that the feature space becomes a line instead of being a circle, but with higher dimensions there is less topological distinction between non-negative and unconstrained).

Problems in the maths:
- The proof of Theorem 2 appears to be incomplete. In the hardmax case, the given proof says that for any s_1 that is close to c_1, there is a better position on the geodesic between c_2 and c_3. This means that switching to that point will get better reward than staying at s_1, in the absence of player 2. However the whole point is that there is a player 2, who needs to be taken into account. The logic would need to go something like: for any s_1, player 2 will select this particular s_2, which means that player 1 is not playing a best response to player 2. It is quite possible that the theorem is true (I do not have the time to check) but the proof is not correct.
- In the statement of Proposition 1, what is the point of defining \hat{c}? It is a scaling of E(c), to which the first (and only) thing that is done is a further scaling to get \bar{c}.
- In the proof of Proposition 1, I think more care needs to be taken. In the first part, what if there is no such theta_m? For example if there is a theta_m such that P(theta_c >= theta_m) >0.5 and P(theta_c <= theta_m) > 0.5 (i.e. P(theta_c=theta_m) is positive, and it just so happens that this is right in the middle of the distribution). For the non-existence when d>2, same complaint as for the proof of Theorem 2 - you need to complete the argument instead of just showing that s_1 is not a best response to itself. (I’ve just had a thought - are you assuming everything is symmetric, so a pure Nash equilibrium must have all s_i equal? That seems to be an anomalous piece of logic - just because the game is symmetric does not mean that the equilibrium must be symmetric - although is this actually the result that is buried inside the 2nd paragraph of the softmax part of the proof, and does that apply t hardmax too?).
- I have not checked the proofs of Lemmas 1 and 2.
- Proposition 2 is not proved




**Summary Of The Paper:**

This is a very cool paper, considering the strategic incentives of multiple content creators when producing content for a YouTube-like recommender system. The article addresses questions of whether the setup of the recommending algorithm has an effect on the type of aggregate content-producer behaviour we would expect to see. This is an important and interesting question, and my first reaction on reading the paper was that I wanted to hear these people present their work and then sit down and discuss it with them in detail! The key claims are that restricting the recommender system to non-negative domains improves the behaviour of the whole system, and that by the recommender system being more explorative it induces content creators to be more conservative (tackling the central point of the content consumer distribution, instead of specialising).

Unfortunately I am writing this review in something of a rush, having only been asked to do it because others dropped out. It is quite possible that I have missed some details in my reading of it.

**Summary Of The Review:**

A really exciting piece of research, let down by a bit of over claiming, and some sloppiness in the presentation.

---

> ### Author Response · Authors · 2022-11-19
> **Rebuttal**
>
> Thank you for taking the time to read and review our paper!
>
> Please see our responses below; we have started working on the corresponding changes (see uploaded revised version), and will incorporate all of them in the final revision & credit anonymous reviewers in the acknowledgements. If you have any further concerns, we would be keen to address them as well.
>
> ### General comments
>
> * **Re Section 3.2**: We have revisited all our figures and made them much more readable. We have also clarified the definition of the metrics we used for the y-axes of our plots, and added further explanations to previously unexplained components of our plots. Regarding comment about clustering, we have clarified in the text that we define a cluster of producers as a set of points whose euclidean distances from one another are negligible (less than 1e-5$\sqrt{d}$).
> * **Re non-negative to unconstrained feature spaces:** Effects of the non-negativity constraint are studied theoretically in Proposition 1, and empirically within the results section (main point of comparison is `nmf`vs. `pmf`, i.e., matrix factorisation with and without the non-negativity constraint). The experiments show that the differences can be significant even beyond the two-dimensional two-player setting. In our view, the combination of the two is quite interesting, as it highlights an example of a seemingly innocuous algorithmic choice with potentially significant impact on creator incentives (with all the caveats discussed esp. in Section 4). We agree our _theoretical_ understanding in particular is limited to two dimensions. Just as a curved line segment is different from a circle, the intersection of a d-sphere with the non-negative orthant is different from the unconstrained d-sphere (due to the presence of boundaries). The question of how this affects competitive equilibria deserves further study in the future. We will highlight this as an open question in the final revision.
> * **Re reproducibility:** As mentioned above, we will add more details about the experiments, and a README file to the code. We would be happy to address further concerns, if any.
>
>
> ### Maths
>
> Thank you for the detailed comments on our proofs; we have updated the revision to clarify/address all the issues that you mention, as explained below.
>
> * **Proof of Theorem 2**: The argument implies that in a PNE $s$, both players must have $u_i(s) \geq \frac{2}{3}$. This is a contradiction since $\sum_i u_i(s) = 1$ by the definition of the exposure utility. Hence no PNE exists.
> * **Use of $\hat{c}$**: We used $\hat{c}$ as a shorthand since we need it at two places in the main body (the assumption that $\hat{c} \neq 0$, and the definition of $\bar{c}$), and throughout the appendix (proofs). Within the main body, it helps us fit into the strict 9 page limit.
> * **Proposition 1, first part**: We have a typo in the definition of the median $\theta_m$ which should have had _inequalities_ $\mathbb{P}(\theta_m \geq \theta_c) \geq \frac{1}{2}$ and $\mathbb{P}(\theta_m \leq \theta_c) \geq \frac{1}{2}$ (instead of $= \frac{1}{2}$); thanks for spotting this! In the particular case you mention (i.e., $\mathbb{P}(\theta_m = \theta_c) > 0$), the median is unique, and thus $s_1 = s_2 = \theta_m$ is a PNE, as any deviation would yield utility strictly lower than $\frac{1}{2}$.
> * **Proposition 1, non-existence when $d > 2$**: Similarly to the case of Theorem 2 above, the proof implies that any PNE $s$ must have $u_1(s) \geq \frac{2}{3}$ and $u_2(s) \geq \frac{2}{3}$, which is a contradiction.
> * **Proposition 2 not proved**: Thanks for spotting this! The result holds because when $s_1 = \cdots = s_n = \bar{c}$, the gradient from Equation (4) is the same for all producers, and it is proportional to $(I - \bar{c} \bar{c}^\top) \hat{c}$, which is equal to zero by $\bar{c} = \hat{c} / \\| \hat{c} \\|$.

---

> > ### Comment · Reviewer_rzZd · 2022-11-25
> > **Thank you for your rebuttal**
> >
> > Thank you for taking the time to carefully respond to my review. The response, and that to the other reviewers, has satisfied me that my concerns will be suitably addressed in the final version of the paper, and I am amending my score upward.

---

### Official Review · Reviewer_6hAx · 2022-11-04

**Confidence:** 3
**Correctness:** 3
**Technical Novelty And Significance:** 4
**Empirical Novelty And Significance:** 2
**Recommendation:** 8

**Clarity, Quality, Novelty And Reproducibility:**

The paper is mostly clear and well-written. The technical novelty is substantial as dynamics in the recommender system are well formalized and analyzed. The reproducibility is good, but the empirical evaluation and justification may not be comprehensive enough.

**Strength And Weaknesses:**

Strengths:
- The paper is well-written and flows very smoothly. Despite the paper's modest space, it provides the required context and necessary explanation in a good job.
- Most parts of the paper have clear motivations. Many of my questions are well answered in the paper.
- Formulation and theoretical analysis of the exposure game in the recommender system is insightful. Most papers studied the recommender system in a fixed manner. I agree understanding the interaction effects/dynamics between the producer and customer is important and useful.
- The analysis of diversity and exploration is interesting. The paper mentioned the game may concentrate on uniform distribution, and exploration may incentivize content that is uniform and broadly appealing rather than diverse. From the recent papers, I also found deeper models can achieve diversity/coverage and accuracy at the same time. These two concepts may not be contradictory. If could further reveal the relationship between these two, it'll be very interesting.
- The paper identified several factors that can influence the recommendation seriously and provided a pre-deployment audit tool that can benefit the community.

Weaknesses/Questions:
- In eq. (2), the paper introduced the temperature parameter $\tau$ to control the spread of exposure probabilities. Many of the follow-up analyses and experiments are built on this. However, most recommender systems didn't include this in their objectives. Could you please explain the rationale or connectivity between these?
- On page 3, I'm confused about the full control assumption. What's the difference between full contry and partial control and why it can abstract away the explicit model of producer actions? Is it possible to list some concrete examples?
- For experiments, how are the producers defined? In my opinion, there are only user/item information in these data sets.
- For the $\epsilon$-LNE solver in eq. (4), is this the paper originally proposed or adapted from others? I'm not familiar with the NE-based methods. It seems it's also updating the model with the gradient. Could you please illustrate a little more on the common points and difference between this one and the normal gradient descent method used to optimize ML model?

**Summary Of The Paper:**

The paper studied the interaction effects between content producers and consumers. The dynamics in the interactions are well formalized as an exposure game about incentives. They provided a comprehensive theoretical analysis of the properties of Nash equilibria in the games and proposed tools for numerically finding equilibria in exposure games. Both theory and empirical experiments verify the dependence between algorithmic exploration and content diversity, and between model expressivity and bias.

**Summary Of The Review:**

In general, I think it's a good paper with clear novelty. Some of the descriptions and empirical experiments in the paper may require further explanations.

---

> ### Author Response · Authors · 2022-11-19
> **Rebuttal**
>
> Thank you for taking the time to read and review our paper!
>
> Please see our responses below; we have started working on the corresponding changes (see uploaded revised version), and will incorporate all of them in the final revision & credit anonymous reviewers in the acknowledgements. If you have any further concerns, we would be keen to address them as well.
>
>
> ### General comments
>
> * **Re motivation for temperature parameter $\tau$**: The temperature parameter can be understood as a relaxation of top-1 recommendation (corresponding precisely to $\tau=0$). The softmax case ($\tau > 0$) captures nondeterminism, allowing non-zero probabilities of exposure for items with rank larger than 1. This could capture the effect of exploration by recommenders (e.g. the softmax arises from max-entropy strategies), or user behaviour in which lower ranked items receive small amounts of traffic. This model is used in applications like YouTube (see Box 1, or Chen et al., 2019, equation 6). A generalisation of softmax to the top-N setting is the Plackett-Luce model, which is widely used in the literature on recommender fairness [1, 2, 3]. We will include these explanations in the final revision.
> * **Re full control assumption**: By full control, we mean that the producers can place the _embedding of their content_ $s_i$ anywhere in $S^{d-1}$. In reality, producers can only adapt _features of their content_ (e.g., video topic & length on YouTube, or keywords & links in SEO). Because producers typically _don’t_ know the transformation which maps content to embedding, they don’t have full control of where $s_i$ is going to be. On the other hand, producers often have a large variety of tools at their hand, as illustrated, e.g., by the variety of techniques employed in SEO. Since there is significant incentive to achieve high exposure, it is not unrealistic to model producers as being able to find exposure maximising content. The full-control assumption approximates this scenario, without explicitly modelling the producer action space, which is what we mean by “abstracting it away”. We will clarify this in the final revision.
> * **Re producer definition in experiments**: Defining the exposure game only requires embeddings of the _users_  which we fit from rating data using matrix factorization methods (the corresponding item embeddings are not used in the experiments). Independent of the original dataset, we define a number of producers. We focus on _pure_ $\epsilon$-LNE, corresponding to each producer creating a single new item.We will clarify this relationship between producers and items in the final revision.
> * **Re solver in experiments**: We are not the first to use gradient ascent for finding $\epsilon$-LNE, as the same ideas have been present & used, e.g., in [4, 5] . Essentially, the algorithm runs $n$ _independent_ gradient ascent optimizers, each following the gradients of the utility $u_i(s)$. The optimizers execute steps simultaneously, i.e., the iterate at step $t + 1$ is obtained by assuming that the other players play the strategy from step $t$. We will add these references and a description of similarities and differences to our revision.
>
>
> ### Empirical evaluations
>
> * We will add more details about the experiments based on comments from all the reviewers. We would be happy to incorporate further suggestions.
> * The main purpose of Section 3 is to illustrate the potential of our proposed pre-deployment algorithmic audit. This method of auditing based on creator behaviour models is novel. More than just illustrating our theory empirically, Section 3 presents a complementary contribution which motivates the development of creator behaviour models, such as exposure games. We would be happy to incorporate further suggestions on how to better illustrate the audit (or our other results).
>
>
> ### References
>
> [1] Kotary, J., Fioretto, F., Van Hentenryck, P. and Zhu, Z., 2022. End-to-End Learning for Fair Ranking Systems.
> [2] Yadav, H., Du, Z. and Joachims, T., 2021. Policy-gradient training of fair and unbiased ranking functions.
> [3] Singh, A. and Joachims, T., 2019. Policy learning for fairness in ranking.
> [4] Singh, S., Kearns, M.J. and Mansour, Y., 2000. Nash Convergence of Gradient Dynamics in General-Sum Games.
> [5] Balduzzi, D., Racaniere, S., Martens, J., Foerster, J., Tuyls, K. and Graepel, T., 2018. The mechanics of n-player differentiable games.

---

> > ### Comment · Reviewer_6hAx · 2022-11-25
> > **Response to Authors**
> >
> > Thanks very much for your response. I appreciate the time and efforts you put on rebuttal. The response is very clear and well addressed my concerns. Therefore, I raised my score accordingly. Please go ahead to include these discussions and explanations in the future version.

---

### Decision · Program_Chairs · 2023-01-20

**Decision:**

Accept: notable-top-5%

**Justification For Why Not Higher Score:**

N/A

**Justification For Why Not Lower Score:**

As I mentioned above, I don't have enough expertise in game theory to make an informed decision all by myself (reflected by my confidence score below). But I have seldom seen reviewers with genuine excitement about a piece of work and want to have it published so that they can have a discussion with the authors (normally you see the opposite). For this I think this paper deserves to be an oral presentation.

**Metareview: Summary, Strengths And Weaknesses:**

Summary: This paper considers the problem that in a recommender system, the interaction between content creators and users (content creators compete for limited user attention) can be formulated as a so-called exposure game about incentives. The paper provides a formal treatment characterizing the properties of Nash equilibria in the games and point out that how some of the algorithmic choices can change and affect such equilibria. The paper presents both theoretical and empirical evidence.

Strengths: All the reviewers unanimously agree that this is a very solid contribution with very interesting implications. The analysis is rigorous and insightful. Being a theoretical paper, it is also well-written and easy to follow. I have to admit game theory is not in my expertise. However, all three reviewers are praising this paper with genuine excitement and I think that is quite telling.

Weaknesses: I don't think there are any major weaknesses -- some of the minor weaknesses around the proofs and confusion around the clarity have been addressed during the discussion phase. The authors should take the feedback into account when preparing for the final version of the paper.



**Note From Pc:**

if the above contains the word "oral" or "spotlight" please see: "oral" presentation means -> notable-top-5% and "spotlight" means -> notable-top-25%. As stated in our emails, we are disassociating presentation type from AC recommendations